# Improvement of islet transplantation by the fusion of islet cells with functional blood vessels

Lisa Nalbach[1], Leticia P Roma[2], Beate M Schmitt[1], Vivien Becker[1], Christina Körbel[1], Selina Wrublewsky[1], Mandy Pack[1], Thomas Später[1], Wolfgang Metzger[3], Maximilian M Menger[1,4], Florian S Frueh[5], Claudia Götz[6], Haopeng Lin[7], Joseline EM Fox[7], Patrick E MacDonald[7], Michael D Menger[1], Matthias W Laschke[1] iD & Emmanuel Ampofo[1,*] iD

## Abstract

Pancreatic islet transplantation still represents a promising therapeutic strategy for curative treatment of type 1 diabetes mellitus. However, a limited number of organ donors and insufficient vascularization with islet engraftment failure restrict the successful transfer of this approach into clinical practice. To overcome these problems, we herein introduce a novel strategy for the generation of prevascularized islet organoids by the fusion of pancreatic islet cells with functional native microvessels. These insulin-secreting organoids exhibit a significantly higher angiogenic activity compared to freshly isolated islets, cultured islets, and non-prevascularized islet organoids. This is caused by paracrine signaling between the β-cells and the microvessels, mediated by insulin binding to its corresponding receptor on endothelial cells. *In vivo*, the prevascularized islet organoids are rapidly blood-perfused after transplantation by the interconnection of their autochthonous microvasculature with surrounding blood vessels. As a consequence, a lower number of islet grafts are required to restore normoglycemia in diabetic mice. Thus, prevascularized islet organoids may be used to improve the success rates of clinical islet transplantation.

**Keywords** diabetes; insulin; islet transplantation; microvascular fragments; vascularization
**Subject Categories** Metabolism; Vascular Biology & Angiogenesis

## Introduction

The development of the Edmonton protocol, a glucocorticoid-free immunosuppressive regimen, represents a milestone in clinical pancreatic islet transplantation (Shapiro *et al*, 2000). Since then, the rate of islet transplantation activity has increased all over the world (Shapiro *et al*, 2003). However, multiple donors are still required to achieve insulin independence. This is mainly due to the fact that the isolation procedure strips the dense microvascular network of islets (Brissova & Powers, 2008), leading to their failed engraftment after transplantation (Gamble *et al*, 2018).

Significant research efforts have been conducted to promote the revascularization and, thus, to improve nutrient and oxygen supply of grafted islets during the first post-transplantational days (Song *et al*, 2019; Menger *et al*, 2020). A promising strategy to achieve this is the prevascularization of the transplantation site, as already described for subcutaneous tissue by means of a catheter technique (Pepper *et al*, 2015). However, in this case the revascularization of the islet grafts is crucially dependent on the angiogenic ingrowth of new microvessels from the surrounding host tissue, which is still a time-consuming process. Another promising strategy is the incorporation or co-transplantation of endothelial cells (Song *et al*, 2010; Pan *et al*, 2011; Oh *et al*, 2013; Barba-Gutierrez *et al*, 2016). However, the isolation of primary endothelial cells out of different tissues (Hewett, 2016) or the generation of pluripotent stem cell-derived endothelial cells (Ikuno *et al*, 2017) are not feasible approaches in daily clinical practice, yet. Moreover, the development of new blood vessels originating from single endothelial cells requires up to 10 days (Laschke & Menger, 2016), which is not fast enough to guarantee the survival of islets in the initial phase after transplantation. To overcome this problem, we herein introduce a novel strategy for the generation of prevascularized islet organoids by the fusion of pancreatic islet cells with microvascular fragments (MVF).

1 Institute for Clinical & Experimental Surgery, Saarland University, Homburg/Saar, Germany
2 Biophysics Department, Center for Human and Molecular Biology, Saarland University, Homburg/Saar, Germany
3 Department of Trauma, Hand and Reconstructive Surgery, Saarland University, Homburg, Germany
4 Departement of Trauma and Reconstructive Surgery, Eberhard Karls University Tuebingen, Tuebingen, Germany
5 Department of Plastic Surgery and Hand Surgery, University Hospital Zurich, University of Zurich, Zurich, Switzerland
6 Medical Biochemistry and Molecular Biology, Saarland University, Homburg, Germany
7 Department of Pharmacology, Alberta Diabetes Institute, University of Alberta, Edmonton, AB, Canada
*Corresponding author. Tel: +49 6841 16 26561; Fax: +49 6841 16 26553; E-mail: emmanuel.ampofo@uks.eu

Microvascular fragments are functional segments of arterioles, capillaries, and venules with a length of 20–180 µm, which can be enzymatically isolated in large amounts from adipose tissue within a short period of time (Hoying *et al*, 1996; Pilia *et al*, 2014; Laschke & Menger, 2015). They still exhibit the physiological morphology of native microvessels, consisting of an endothelial cell-lined lumen covered by stabilizing α-smooth muscle actin (SMA)-positive cells (Hoying *et al*, 1996). Moreover, MVF release numerous angiogenic growth factors, including vascular endothelial growth factor (VEGF) and basic fibroblast growth factor (bFGF) (Pilia *et al*, 2014). They are characterized by a high proliferative activity and rapidly reassemble into blood-perfused microvascular networks by interconnecting with each other and the surrounding microvasculature after transplantation (McDaniel *et al*, 2014). Accordingly, MVF have been suggested as potent vascularization units for tissue engineering and regenerative medicine (Laschke & Menger, 2015; Später *et al*, 2017; Frueh *et al*, 2017a; Frueh *et al*, 2017b).

Based on these findings, we speculated that the incorporation of MVF into islet organoids markedly improves the constructs' *in vivo* vascularization capacity. To prove our hypothesis, we fused MVF with mouse or human pancreatic islet cells to generate prevascularized islet organoids (PI + MVF). These PI + MVF showed a markedly improved *in vivo* vascularization, viability, and functionality when compared to controls. Hence, the present proof-of-principle study introduces a highly promising tissue engineering concept for the treatment of type 1 diabetes (T1D).

# Results

### Generation of prevascularized islet organoids

We isolated islets from donor mouse pancreata by collagenase digestion and further dissociated these islets into single cells by a second enzymatic digestion step (Fig 1A). In addition, we harvested the epididymal fat pads from male mice and processed the adipose tissue mechanically and enzymatically to obtain a mixture of MVF and extracellular matrix (ECM) (Fig 1B). This mixture was then filtrated to gain purified individual MVF (Fig 1B). We next characterized MVF by means of Western blot analysis using murine and human endothelial cell lines as controls. As expected, MVF expressed the endothelial cell marker CD31 (Fig 1C). In contrast to the endothelial cells, they also strongly expressed the pericytic marker protein α-SMA (Fig 1C). In line with these findings, immunohistochemical analyses demonstrated that MVF still exhibit

the physiological morphology of native blood vessels, consisting of an endothelium covered by a perivascular layer of α-SMA- and platelet-derived growth factor receptor (PDGFR)-β-positive pericytes (Fig 1D).

In a next step, we applied the liquid overlay technique to generate prevascularized islet organoids (Fig 1E). For this purpose, we tested different time periods and ratios between MVF and MIN6 β-cells to establish the ideal protocol for the formation of stable tissue constructs of standardized size and shape. This was achieved by the cocultivation of 2,000 MIN6 β-cells with 200 MVF for 5 days in liquid overlay culture (Fig EV1A). Based on these promising results, we generated PI + MVF out of primary pancreatic islet cells from donor mice (Fig 1F). Pseudo-islets (PI) consisting of 2,000 islet cells as well as freshly isolated islets (FI) and islets cultivated for 5 days (CI) served as controls. FI, CI, and PI exhibited a comparable diameter of ~ 160–170 µm (Fig EV1B). PI + MVF presented with a significantly larger diameter of ~ 210 µm due to the incorporation of MVF (Fig EV1B). Morphological scanning electron microscopic analysis illustrated that PI + MVF are characterized by a heterogeneous surface pattern, whereas the surface of PI and CI appears smooth (Fig 1G). The surface of FI was rather rough (Fig 1G), most probably due to the fact that they were directly analyzed after the enzymatic isolation procedure, which may have caused disruption of the outer cell layers.

### Viability of prevascularized islet organoids

To assess whether the cells within PI + MVF were viable, we performed neutral red and trypan blue stainings. We did not detect any differences between PI + MVF, FI, CI, and PI (Figs 2A and EV1C). More detailed flow cytometric analysis of propidium iodide/annexin V-stained cells revealed a significantly higher number of vital cells in PI + MVF ($92 \pm 4\%$) when compared to CI ($79 \pm 1\%$) and PI ($72 \pm 4\%$) (Fig 2B and C), indicating that MVF may have beneficial effects on islet cell survival.

Long-term cultivation of islets for more than 2 days is known to induce hypoxic stress (Giuliani *et al*, 2005), which leads to elevated intracellular reactive oxygen species (ROS) levels (Ma *et al*, 2014) and reduced β-cell viability (Cantley *et al*, 2010). To examine whether MVF are able to lower ROS levels within islet organoids, we next used islets from transgenic mice expressing the mitochondrial $H_2O_2$ biosensor mito-roGFP2-Orp1 (Fujikawa *et al*, 2016) for PI + MVF generation. This biosensor is radiometric and changes its fluorescence emission according to the ratio of oxidized (405 nm) to reduced roGFP2 (488 nm), which depends on mitochondrial $H_2O_2$

---

**Figure 1. Generation of PI + MVF.**

A  A neutral red-stained mouse pancreas is excised and digested with collagenase. Islets are then separated by hand picking and further digested into single islet cells. Scale bars: 10 mm (left image); 200 µm (middle image); 60 µm (right image).

B  Epididymal fat pads are isolated and processed mechanically and enzymatically. The obtained mixture, containing MVF and ECM, is filtrated for the purification of MVF. Scale bars: 20 mm (left image); 25 µm (right image).

C  Western blots of CD31, α-SMA, and GAPDH in whole cell extracts of eEND, HDMEC, HUVEC, and MVF.

D  Immunofluorescence stainings of CD31/α-SMA and CD31/PDGFR-β in MVF. Cell nuclei are stained with Hoechst 33342 (blue). Scale bar: 20 µm.

E  Schematic illustration of PI + MVF generation. By means of liquid overlay technique, 2,000 pancreatic islet cells and 200 MVF are cocultured for 5 days.

F  PI and PI + MVF on days −5, −3, and 0. Scale bar: 200 µm.

G  Surface morphology of FI, CI, PI, and PI + MVF on day 0 as visualized by scanning electron microscopy. Scale bar: 50 µm.

Source data are available online for this figure.

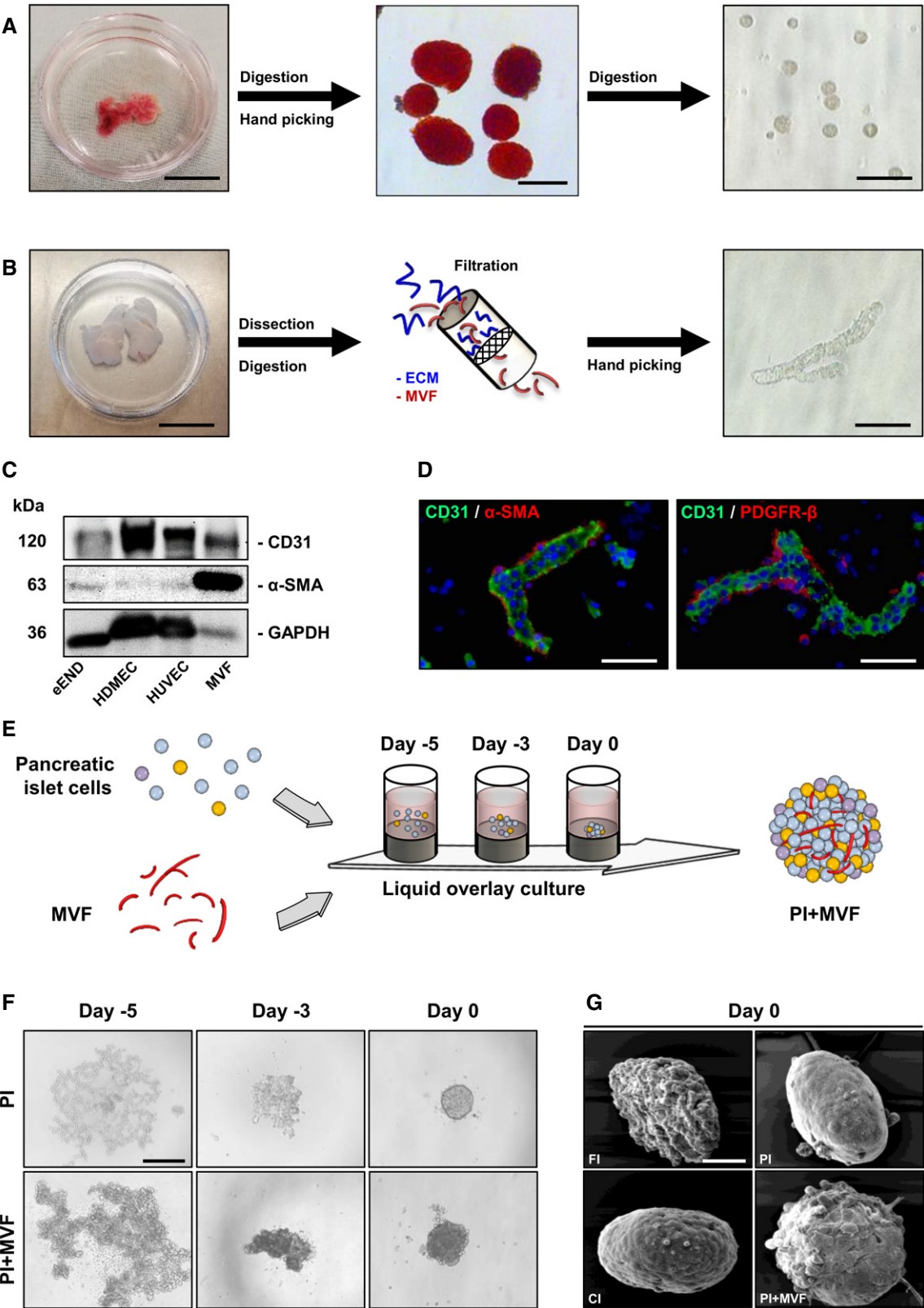

**Figure 1.**

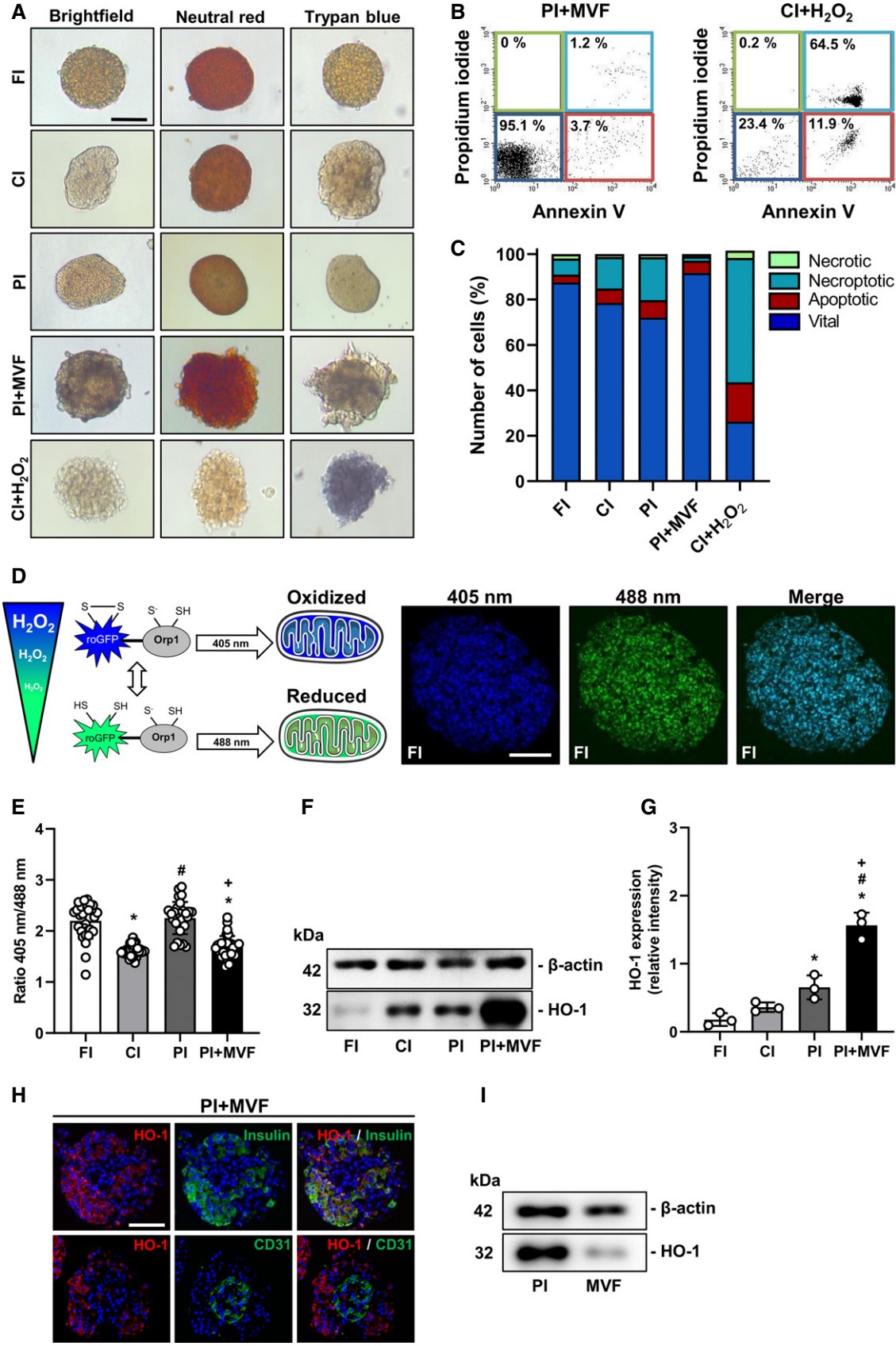

**Figure 2.**

**Figure 2. Viability of FI, CI, PI, and PI + MVF.**

A   Neutral red and trypan blue stainings of FI, CI, PI, and PI + MVF. CI incubated for 24 h with 0.2% $H_2O_2$ served as control. Scale bar: 100 μm.
B   Representative flow cytometric scatterplots of propidium iodide/annexin V-stained cells from PI + MVF and CI + $H_2O_2$ (positive control).
C   Quantitative analysis of propidium iodide/annexin V-stained cells from FI, CI, PI, PI + MVF, and CI + $H_2O_2$ subdivided in necrotic, necroptotic, apoptotic, and vital cells in % of total cell number (n = 3 each).
D   Schematic illustration of the mitochondrial ROS sensor mito-roGFP2-Orp1. The fluorescence maximum of this sensor shifts from 488 to 405 nm with increasing mitochondrial $H_2O_2$ levels, as shown by representative fluorescence images of FI. Scale bar: 70 μm.
E   Quantitative analysis of the ROS sensor ratio (405 nm/488 nm) of FI, CI, PI, and PI + MVF (n = 25 each). Mean ± SD. One-way ANOVA and Tukey's multiple comparisons post hoc test were used for statistical analysis. *$P < 0.05$ vs. FI; #$P < 0.05$ vs. CI; +$P < 0.05$ vs. PI.
F   Representative Western blot analysis of β-actin and HO-1 in whole cell extracts from FI, CI, PI, and PI + MVF.
G   Quantitative analysis of HO-1 expression in FI, CI, PI, and PI + MVF (n = 3 each). Mean ± SD. One-way ANOVA and Tukey's multiple comparisons post hoc test were used for statistical analysis. *$P < 0.05$ vs. FI; #$P < 0.05$ vs. CI; +$P < 0.05$ vs. PI.
H   Representative immunofluorescence stainings of HO-1/insulin and HO-1/CD31 in PI + MVF. Cell nuclei are stained with Hoechst 33342 (blue). Scale bar: 50 μm.
I   Representative Western blot analysis of β-actin and HO-1 in whole cell extracts from PI and MVF.

Data information: Exact P-values are specified in Appendix Table S1.

levels (Fig 2D). In contrast to Giuliani and colleagues (Giuliani *et al*, 2005), we detected reduced ROS levels in CI when compared to FI (Fig 2E), which may be explained by increased cellular stress during the isolation procedure. Moreover, ROS levels in PI were higher when compared to CI, indicating that the reaggregation process during the generation of islet organoids rather than the cultivation itself triggers ROS formation (Fig 2E). Of interest, MVF were capable of protecting against this reaggregation-induced ROS formation (Fig 2E). Additional Western blot analysis demonstrated that the incorporation of MVF into islet organoids markedly increases the expression of heme oxygenase (HO)-1 (Fig 2F and G), which is one of the major ROS-scavenger enzymes in pancreatic β-cells (Pileggi *et al*, 2001). To discriminate if endocrine or endothelial cells or both express HO-1, we performed immunohistochemical analyses of PI + MVF showing that HO-1 expression is mainly restricted to islet cells (Fig 2H). This was verified by additional Western blot analyses of PI and MVF (Fig 2I).

### Cellular composition, endocrine function, and angiogenic activity of prevascularized islet organoids

To investigate the cellular composition of FI, CI, PI, and PI + MVF, we analyzed the fractions of insulin-, glucagon-, and somatostatin-producing cells by means of immunohistochemistry (Fig 3A). In this context, it should be noted that the endocrine cell number may differ between individual islets depending on their size. Moreover, individual MVF also differ in their length and, thus, cellular content. Most MVF used in this study exhibited a length of ~10–60 μm (Fig EV1D). Due to these differences of cell numbers within individual islets and islet organoids, the data of the cellular composition were normalized to the overall number of cells within FI, CI, PI, and PI + MVF per section. Our quantitative analyses revealed a significantly lower fraction of insulin-positive cells in CI, PI, and, in particular, in PI + MVF when compared to FI (Fig 3B). The latter finding can be explained by the fact that the additional incorporation of MVF markedly increases the overall cell number of the organoids and, thus, reduces the fraction of insulin-positive cells. On the other hand, we detected a lower fraction of glucagon-positive cells in FI when compared to all cultivated groups (Fig 3C), which may be due to the secretion of glucagon in low glucose-containing medium during the hand picking procedure. We further determined a significantly lower fraction of somatostatin-positive cells in PI + MVF (Fig 3D).

In addition, we analyzed the expression and secretion of insulin in FI, CI, PI, and PI + MVF by means of Western blot and enzyme-linked immunosorbent assay (ELISA). We found that the prevascularization of islet organoids with MVF does not affect the absolute expression of the hormone (Fig 3E). However, the normalized glucose-stimulated insulin secretion of PI + MVF was diminished when compared to PI, most probably due to the incorporation of MVF (Fig 3F). In contrast, the absolute insulin secretion and the stimulatory index (SI) did not differ between PI and PI + MVF (Fig EV1E and F).

We next investigated the microvasculature of islets and islet organoids by immunohistological staining of CD31 (Fig 4A and EV1G). We detected ~ 10% CD31-positive cells in FI (Fig 4B). In line with previous studies reporting the progressive loss of endothelial cells during islet cultivation (Parr *et al*, 1980; Mendola *et al*, 1994), both CI and PI exhibited a significantly lower fraction of CD31-positive cells (Fig 4B). As expected, PI + MVF presented with a markedly higher fraction of endothelial cells (Fig 4A and B, and EV1H), as confirmed by additional Western blot analysis (Figs 4C and D, and EV1I). Notably, these cells were clearly arranged in a tube-like distribution pattern, corresponding to the incorporated MVF within the constructs (Fig 4A). Three-dimensional confocal microscopy of the prevascularized islet organoids further showed the organization of these native microvessels into dense microvascular networks in the center of the constructs and their surface (Fig 4E). It should be noted that PI + MVF are not only composed of endocrine and endothelial cells (Fig EV2A). This is because MVF are fully functional native microvessels, which are covered with stabilizing α-SMA-positive pericytes (Figs 1D, and EV2B). In addition, they also contain adipose stromal cells (ASC), a population of pluripotent mesenchymal cells that is predominantly associated with blood vessels (Traktuev *et al*, 2008). Accordingly, we detected adipogenic differentiation in PI + MVF exposed to insulin for 5 days (Fig EV2C).

Insulin binds to insulin receptor (IR) and insulin-like growth factor receptor (IGFR) on endothelial cells, which stimulates their proliferation (Shrader *et al*, 2009). We observed a high fraction of 68 ± 5% Ki67-positive endothelial cells in PI + MVF (Fig 4F and G). The stimulating effect of insulin on endothelial cell proliferation was verified by inhibition of IR/IGFR activity with the receptor antagonist linsitinib, resulting in a markedly suppressed cell proliferation within PI + MVF (Fig 4H). To elucidate the underlying intracellular signaling induced by insulin, MVF were exposed

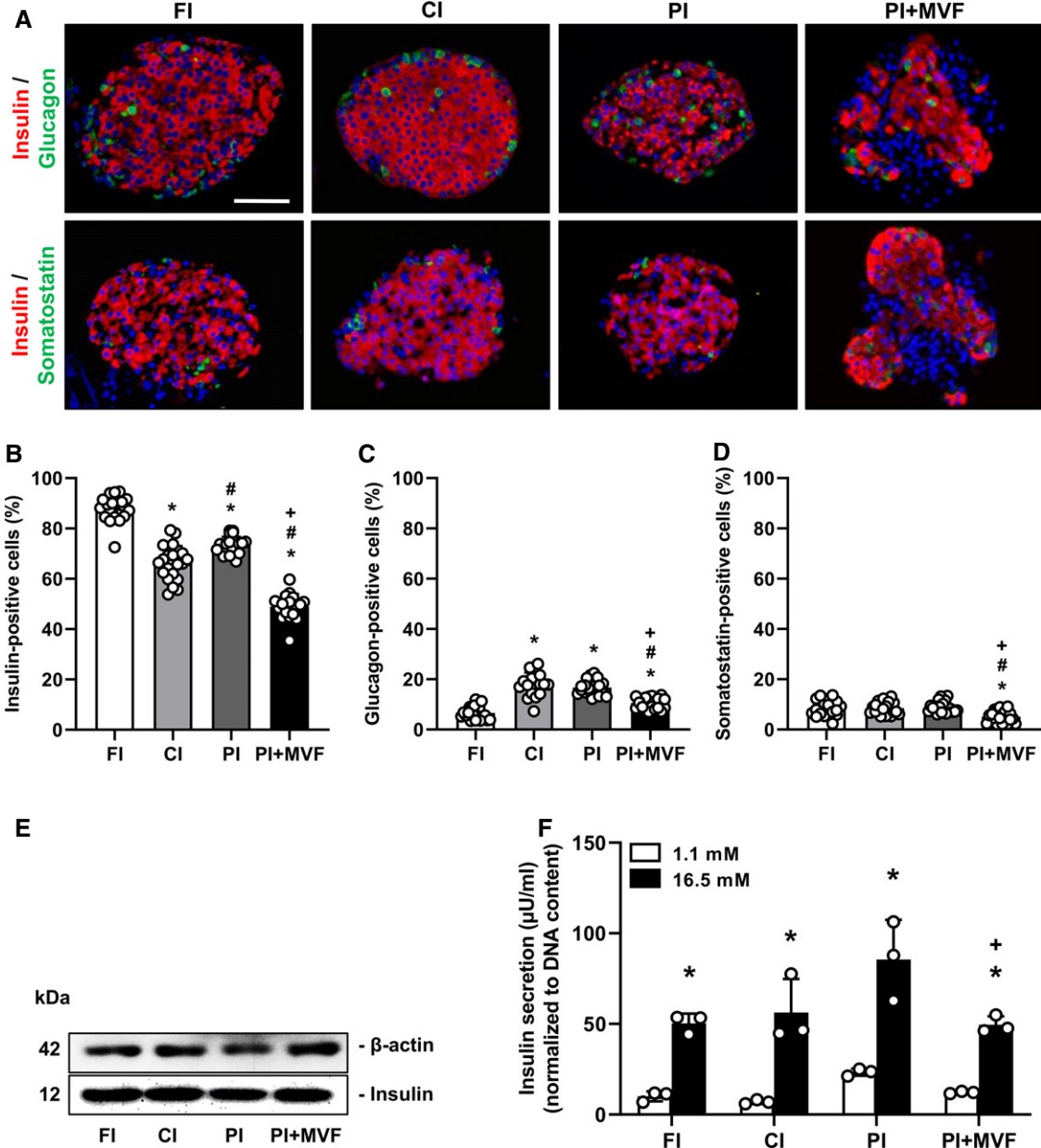

**Figure 3. Endocrine composition and function of FI, CI, PI, and PI + MVF.**

A Representative immunofluorescence stainings of insulin/glucagon and insulin/somatostatin in FI, CI, PI, and PI + MVF. Cell nuclei are stained with Hoechst 33342 (blue). Scale bar: 60 μm.

B–D Quantitative analysis of insulin- (B), glucagon- (C) and somatostatin-positive cells (D) in FI, CI, PI, and PI + MVF in % of all islet or organoid cells ($n = 20$ each). Mean ± SD. One-way ANOVA and Tukey's multiple comparisons *post hoc* test were used for statistical analysis. *$P < 0.05$ vs. FI; #$P < 0.05$ vs. CI; +$P < 0.05$ vs. PI.

E Western blot analysis of β-actin and insulin in whole cell extracts from FI, CI, PI, and PI + MVF.

F Insulin secretion (μU/ml) (normalized to DNA content) of FI, CI, PI, and PI + MVF exposed to 1.1 mM and 16.5 mM glucose ($n = 3$ each). Mean ± SD. One-way ANOVA and Tukey's multiple comparisons *post hoc* test as well as an unpaired Student's *t*-test were used for statistical analysis. *$P < 0.05$ vs. insulin secretion at 1.1 mM glucose within each individual group; +$P < 0.05$ vs. insulin secretion of PI at 16.5 mM glucose.

Data information: Exact *P*-values are specified in Appendix Table S1.
Source data are available online for this figure.

to insulin and the phosphorylation of extracellular-signal regulated kinase (ERK)1/2 and Akt was determined by Western blot (Fig 4I). Quantitative analysis showed increased levels of the two phospho-proteins after insulin treatment (Fig 4J and K).

Moreover, we detected a higher expression of vascular endothelial cell growth factor (VEGF)-A in PI + MVF, which may have also promoted endothelial cell proliferation (Fig EV2D). We next investigated the angiogenic potential of the prevascularized islet

organoids by means of collagen-based sprouting assays (Fig 4L). Our results clearly demonstrate that in contrast to FI, CI, and PI only cells from PI + MVF extensively sprouted into the surrounding matrix (Fig 4M). Additional immunohistochemical analyses revealed that these cells partly stained for CD31, indicating their endothelial nature (Fig 4L). Of note, this sprouting activity could be further improved by exposing PI + MVF to high glucose (Fig EV2E).

## Generation of prevascularized humanized islet organoids

Based on our promising results using murine islets, we next investigated whether it is also possible to generate humanized prevascularized islet organoids. This would represent an important step toward the translation of our novel approach into clinical practice. Human pancreatic islets are isolated and processed in specialized islet isolation centers (Hart & Powers, 2019). After isolation, the islets are

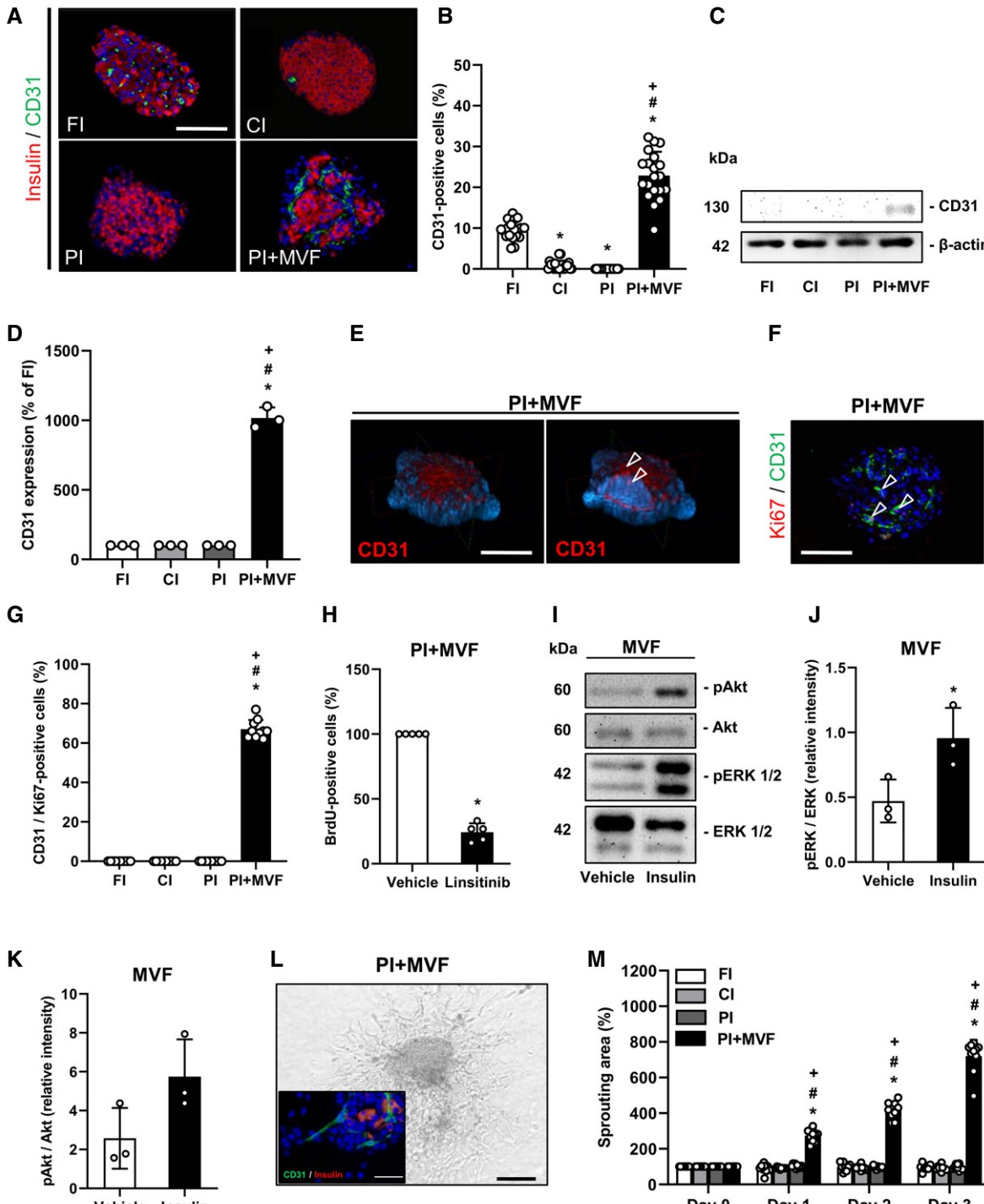

**Figure 4.**

**Figure 4. Microvasculature and angiogenic activity of CI, FI, PI, and PI + MVF.**

A   Representative immunofluorescence stainings of insulin and CD31 in CI, FI, PI, and PI + MVF. Cell nuclei are stained with Hoechst 33342 (blue). Scale bar: 100 μm.

B   Quantitative analysis of CD31-positive cells in FI, CI, PI, and PI + MVF in % of all islet or islet organoid cells ($n = 20$ each). Mean ± SD. One-way ANOVA and Tukey's multiple comparisons *post hoc* test were used for statistical analysis. $^*P < 0.05$ vs. FI; $^\#P < 0.05$ vs. CI; $^+P < 0.05$ vs. PI.

C   Representative Western blot analysis of CD31 and β-actin in whole cell extracts from CI, FI, PI, and PI + MVF.

D   Quantitative analysis of CD31 expression in FI, CI, PI, and PI + MVF in % of FI ($n = 3$). Mean ± SD. One-way ANOVA and Tukey's multiple comparisons *post hoc* test were used for statistical analysis. $^*P < 0.05$ vs. FI; $^\#P < 0.05$ vs. CI; $^+P < 0.05$ vs. PI.

E   Confocal microscopic images of CD31 and cell nuclei (blue) in PI + MVF, demonstrating endothelial cells (arrowheads). Scale bar: 100 μm.

F   Representative immunofluorescence staining of Ki67 and CD31 in PI + MVF. Cell nuclei are stained with Hoechst 33342 (blue). Arrows mark endothelial cells expressing Ki67. Scale bar: 100 μm.

G   Quantitative analysis of CD31/Ki67-positive cells in FI, CI, PI, and PI + MVF in % of all endothelial cells ($n = 10$ each). Mean ± SD. One-way ANOVA and Tukey's multiple comparisons *post hoc* test were used for statistical analysis. $^*P < 0.05$ vs. FI; $^\#P < 0.05$ vs. CI; $^+P < 0.05$ vs. PI.

H   Flow cytometric analysis of BrdU-positive cells from PI + MVF treated with DMSO (vehicle) or linsitinib during the entire fusion process ($n = 5$ each). Mean ± SD. Unpaired Student's *t*-test was used for statistical analysis. $^*P < 0.05$ vs. vehicle.

I   Representative Western blot analysis of pAkt, Akt, pERK1/2, and ERK1/2 from MVF cell extracts. MVF were treated with vehicle or insulin for 3 h.

J, K   Quantitative analysis of pERK1/2/ERK1/2 (J) and pAkt/Akt (K) ratio in PI + MVF treated with vehicle or insulin for 3 h. Data are expressed as relative intensities ($n = 3$ each). Mean ± SD. Unpaired Student's *t*-test was used for statistical analysis. $^*P < 0.05$ vs. vehicle.

L   Representative bright field image of angiogenic sprouting from PI + MVF on day 3 of the sprouting assay. Scale bar: 150 μm. Insert: Immunofluorescence staining of insulin (red) and CD31 (green) of sprouted PI + MVF. Cell nuclei are stained with Hoechst 33342 (blue). Scale bar: 30 μm.

M   Quantitative analysis of sprouting area of FI, CI, PI, and PI + MVF in % of the initial size (day 0) ($n = 10$ each). Mean ± SD. One-way ANOVA and Tukey's multiple comparisons *post hoc* test were used for statistical analysis. $^*P < 0.05$ vs. FI; $^\#P < 0.05$ vs. CI; $^+P < 0.05$ vs. PI.

Data information: Exact *P*-values are specified in Appendix Table S1.
Source data are available online for this figure.

cultivated for 16–35 h before being distributed, which may require an additional 3–5 day-period depending on the transportation distance. As described above, the formation of prevascularized islet organoids takes additional 5 days in liquid overlay culture (Fig 5A). Given these circumstances, we could still show a successful incorporation of murine MVF into human PI (hPI) (Fig 5B). Freshly delivered human islets (hFI) as well as delivered human islets, which were cultivated for additional 5 days (hCI), served as controls (Fig 5B). hFI, hCI, and hPI exhibited a comparable diameter of ~ 165–180 μm, whereas hPI + MVF presented with a significantly larger diameter of ~ 210 μm (Fig EV3A). Immunohistochemical stainings confirmed the fusion of murine MVF and human endocrine islet cells into prevascularized islet organoids, which contained a significantly higher fraction of CD31-positive murine endothelial cells when compared to the other groups (Fig 5C–G). As expected, we additionally detected a substantial number of human endothelial cells in hFI (Fig EV3B and C). Moreover, hPI + MVF were characterized by a similar vascularization pattern when compared to murine islet organoids (Fig 4A and 5C). Accordingly, the humanized islet organoids also exhibited a strong angiogenic activity, as indicated by a significantly increased sprouting area in collagen-based sprouting assays when compared to controls (Fig 5H and I). Of note, the incorporation of MVF into hPI did not affect C-peptide secretion and C-peptide SI, which was comparable to that of hFI (Figs 5J, and EV3D and E).

### *In vivo* vascularization of prevascularized murine islet organoids

In a next set of experiments, we analyzed the *in vivo* vascularization capacity of prevascularized islet organoids in the dorsal skinfold chamber model (Fig 6A–H). For this purpose, we transplanted islets and islet organoids of mouse origin onto the host striated muscle tissue within dorsal skinfold chambers of syngeneic recipient mice (Fig 6A and B). This approach guaranteed a physiological angiogenic response to the grafts, which would have been markedly affected when transplanting humanized islet organoids into immunocompromised recipient animals. Using the dorsal skinfold chamber model, we could repetitively analyze the vascularization of the grafts on days 0 (i.e., day of transplantation), 3, 6, 10, and 14 by means of intravital fluorescence microscopy (Fig 6A and D). This technique did not only allow the quantitative assessment of microvascular network morphology but also the measurement of microhemodynamic parameters. The incorporation of MVF into PI improved the take rate of the grafts, that is, the fraction of engrafted islets or islet organoids in relation to the number of transplants per group on day 14 (Fig 6C). In line with this result, we also found a markedly accelerated and improved vascularization of PI + MVF over time when compared to the other groups. In fact, application of the blood plasma marker 5 % fluorescein isothiocyanate (FITC)-labeled dextran revealed a significantly higher functional capillary density of PI + MVF throughout the entire 14-day observation period when compared to all control groups (Fig 6D and E). Moreover, they exhibited an increased revascularized area (Fig 6F). The measurement of additional microhemodynamic parameters showed that the blood-perfused microvessels within PI + MVF have also a significantly larger diameter as well as a higher centerline red blood cell velocity and volumetric blood flow on day 3 after transplantation (Fig EV4A–C). This markedly enhanced blood perfusion of the islet organoids indicates that their rapid vascularization during the early post-transplant period is rather driven by the interconnection of MVF with the surrounding microvasculature than by the stepwise angiogenic ingrowth of newly formed microvessels from the host tissue, as it was the case for FI, CI, and PI.

We additionally measured the rhodamine 6G-positive area within the grafts as a marker for blood-perfused and vital endocrine cells. This fluorescent dye crosses the endothelial lining within islets and, thus, stains the vessel-surrounding tissue (Diez *et al*, 2017). Of interest, we found in line with our previous results a significantly higher rhodamine 6G-positive area within PI + MVF throughout the entire 14-day observation period when compared to controls (Fig 6G and H).

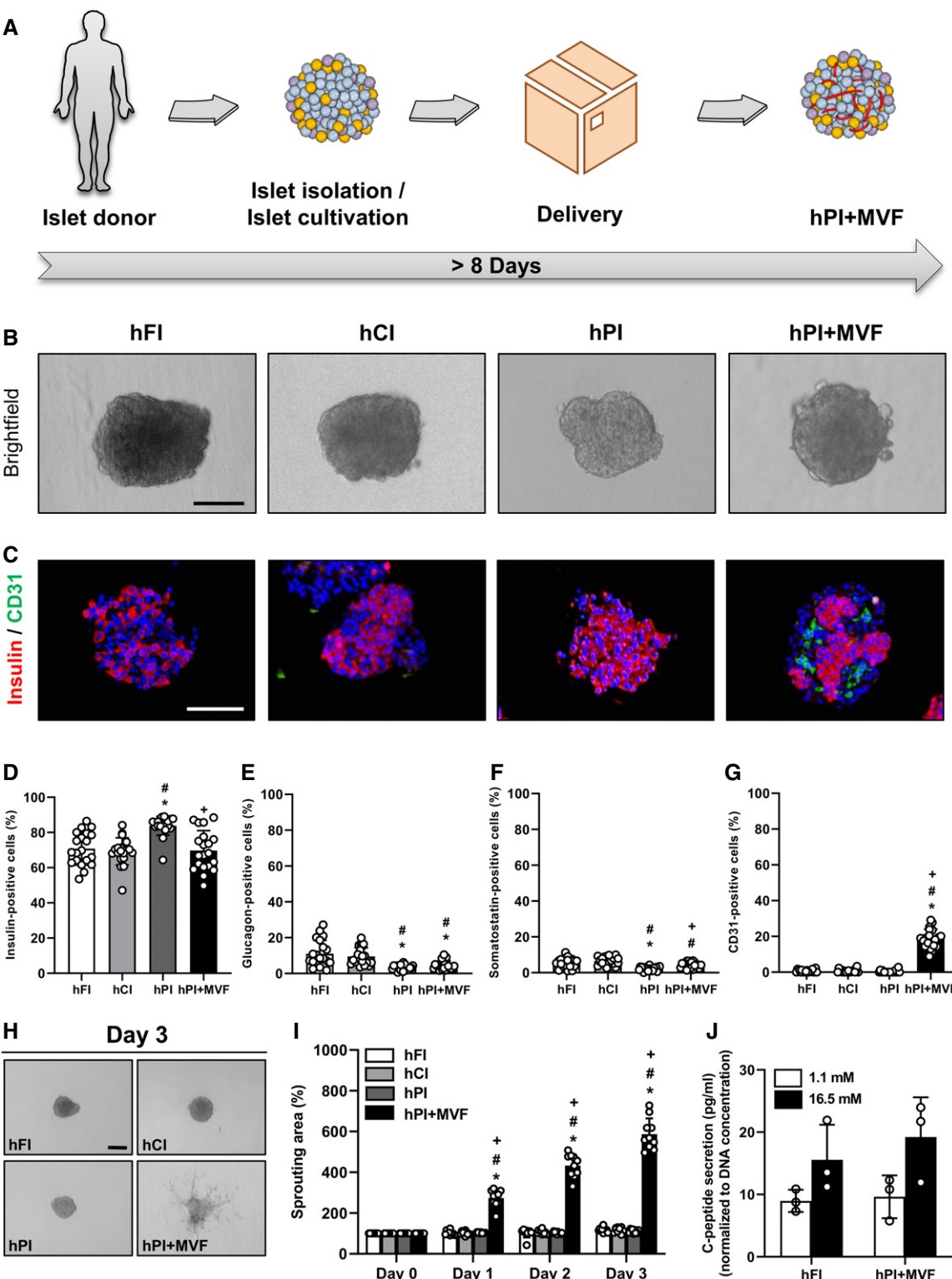

**Figure 5.**

Finally, we analyzed the expression of insulin, glucagon, somatostatin, and CD31 by means of immunohistochemistry to characterize the grafts' cellular composition on day 14 after islet transplantation (Fig 7A). We did not observe any differences in the relative fractions of endocrine cells between all groups (Fig 7B–D). However, as expected, PI + MVF presented with a significantly higher fraction of intra-islet endothelial cells when compared to controls (Fig 7E). To verify that MVF contribute to this high number of intra-islet

**Figure 5. Generation of hPI + MVF.**

A    Schematic illustration of human islet isolation and delivery. Islets are isolated from human donors and cultivated for up to 35 h until they are delivered for the generation of hPI + MVF. The entire process takes more than 8 days.

B    Bright field images of hFI, hCI, hPI, and hPI + MVF. Scale bar: 100 μm.

C    Representative immunohistochemical stainings of insulin and CD31 in hFI, hCI, hPI, and hPI + MVF. Cell nuclei are stained with Hoechst 33342 (blue). Scale bar: 100 μm.

D–G  Quantitative analysis of insulin-(D), glucagon-(E), somatostatin-(F) and CD31-positive cells (G) in hFI, hCI, hPI, and hPI + MVF in % of all islet or islet organoid cells ($n = 20$ each). Mean ± SD. One-way ANOVA and Tukey's multiple comparisons *post hoc* test were used for statistical analysis. *$P < 0.05$ vs. hFI; #$P < 0.05$ vs. hCI; +$P < 0.05$ vs. hPI.

H    Representative bright field images of angiogenic sprouting of hFI, hCI, hPI, and hPI + MVF on day 3 of the sprouting assay. Scale bar: 150 μm.

I    Quantitative analysis of sprouting area of hFI, hCI, hPI, and hPI + MVF in % of initial size (day 0; $n = 10$ each). Mean ± SD. One-way ANOVA and Tukey's multiple comparisons *post hoc* test were used for statistical analysis. *$P < 0.05$ vs. hFI; #$P < 0.05$ vs. hCI; +$P < 0.05$ vs. hPI.

J    Quantitative analysis of C-peptide secretion (pg/ml; normalized to DNA concentration) of hFI and hPI + MVF exposed to 1.1 mM and 16.5 mM glucose ($n = 3$ each). Mean ± SD.

Data information: Exact *P*-values are specified in Appendix Table S1.

Source data are available online for this figure.

endothelial cells, we transplanted additional islet organoids consisting of GFP-positive MVF and GFP-negative islet cells into the dorsal skin-fold chamber of GFP-negative recipient mice. The immunohistochemical analysis of these grafts on day 14 revealed that 34 ± 13% of the CD31-positive endothelial cells within the grafts stained also positive for GFP (Fig EV4D). This indicates that a substantial fraction of the microvasculature within the grafts originated from the incorporated MVF.

**In vivo endocrine function of prevascularized islet organoids**

Since we could show that prevascularized islet organoids exhibit a strong angiogenic and vascularization capacity *in vitro* and *in vivo*, we analyzed whether this may also result in an improved restoration of normoglycemia in streptozotocin (STZ)-induced diabetic animals. Sham-transplanted mice were used to demonstrate the cytotoxic effect of STZ (Fig EV5A–F). STZ-treated mice transplanted with FI served as positive control, whereas nondiabetic animals served as negative controls. In order to define the critical number of FI, which do not restore normoglycemia within 28 days, we transplanted 200, 250, 300, and 400 FI under the kidney capsule of diabetic animals (Fig EV5G and H). Based on these results, we transplanted a critical mass of 250 FI or 250 PI + MVF under the kidney capsule of diabetic mice (Fig 8A). The

body weights did not significantly differ between the groups over the entire observation period (Fig 8B and C). Of note, in the group of PI + MVF-transplanted mice we detected blood glucose levels of 204 ± 80 mg/dl already 4 days after transplantation (Fig 8D). Moreover, normoglycemia was completely restored after 10 days (Fig 8D and E). In contrast, the transplantation of 250 FI did not restore normoglycemia (Fig 8D and E). An intraperitoneal glucose tolerance test (IPGTT) demonstrated that the blood glucose levels of mice transplanted with PI + MVF were lower when compared to animals transplanted with FI, resulting in a significantly reduced AUC (Fig 8F and G). In contrast, the plasma insulin levels, which were determined at 180 min after glucose injection, did not differ between the groups (Fig 8H). At the end of the *in vivo* experiments, the kidneys were harvested and the grafts' cellular composition was analyzed by means of immuno-histochemistry (Fig 8I–K). This analysis revealed no differences in the fraction of insulin-, glucagon-, somatostatin-, and CD31-expressing cells between the two groups (Fig 8J). However, we found a significantly higher overall number of islet cells per high power field (HPF) in mice transplanted with PI + MVF when compared to those receiving FI (Fig 8K). To further clarify whether this was caused by an increased cell proliferation or the inhibition of apoptosis, we analyzed Ki67 and cleaved caspase-3 (casp-3) stainings of the grafts (Fig EV5I). Of interest, we found a

**Figure 6. In vivo vascularization of CI, FI, PI, and PI + MVF.**

A    Schematic illustration of the experimental setting: dorsal skinfold chambers are implanted on day −2 followed by transplantation of FI, CI, PI, and PI + MVF on day 0. Intravital fluorescence microscopy is performed on days 0, 3, 6, 10, and 14 after islet transplantation. On day 14, the tissue is harvested for immunohistochemical stainings.

B    Observation window of a dorsal skinfold chamber with eight neutral red-stained transplanted FI on day 0. Scale bar: 5 mm.

C    Take rate (% of transplanted islets) on day 14 after islet transplantation into the dorsal skinfold chamber ($n = 8$ each). Mean ± SD. One-way ANOVA and Tukey's multiple comparisons *post hoc* test were used for statistical analysis. +$P < 0.05$ vs. PI.

D    Representative intravital fluorescent microscopic images of FI, CI, PI, and PI + MVF within the dorsal skinfold chamber on day 0 and 14. FITC-labeled dextran is used for the visualization of blood-perfused microvessels. Grafts are marked by broken lines. Scale bar: 150 μm.

E, F  Quantitative analysis of the functional capillary density (cm/cm²) (E) and revascularized area (mm²) (F) of transplanted FI, CI, PI, and PI + MVF on day 0, 3, 6, 10, and 14 ($n = 8$ each). Mean ± SD. One-way ANOVA and Tukey's multiple comparisons *post hoc* test were used for statistical analysis. *$P < 0.05$ vs. FI; #$P < 0.05$ vs. CI; +$P < 0.05$ vs. PI.

G    Representative intravital fluorescent microscopic images of FI, CI, PI, and PI + MVF within the dorsal skinfold chamber on day 14. Rhodamine 6G is used to visualize blood-perfused vital endocrine cells (bright signals). Scale bar: 150 μm.

H    Quantitative analysis of the rhodamine 6G-positive area (% of islet size) within transplanted FI, CI, PI, and PI + MVF on day 0, 3, 6, 10, and 14. Mean ± SD. One-way ANOVA and Tukey's multiple comparisons *post hoc* test were used for statistical analysis. *$P < 0.05$ vs. FI; #$P < 0.05$ vs. CI; +$P < 0.05$ vs. PI.

Data information: Exact *P*-values are specified in Appendix Table S1.

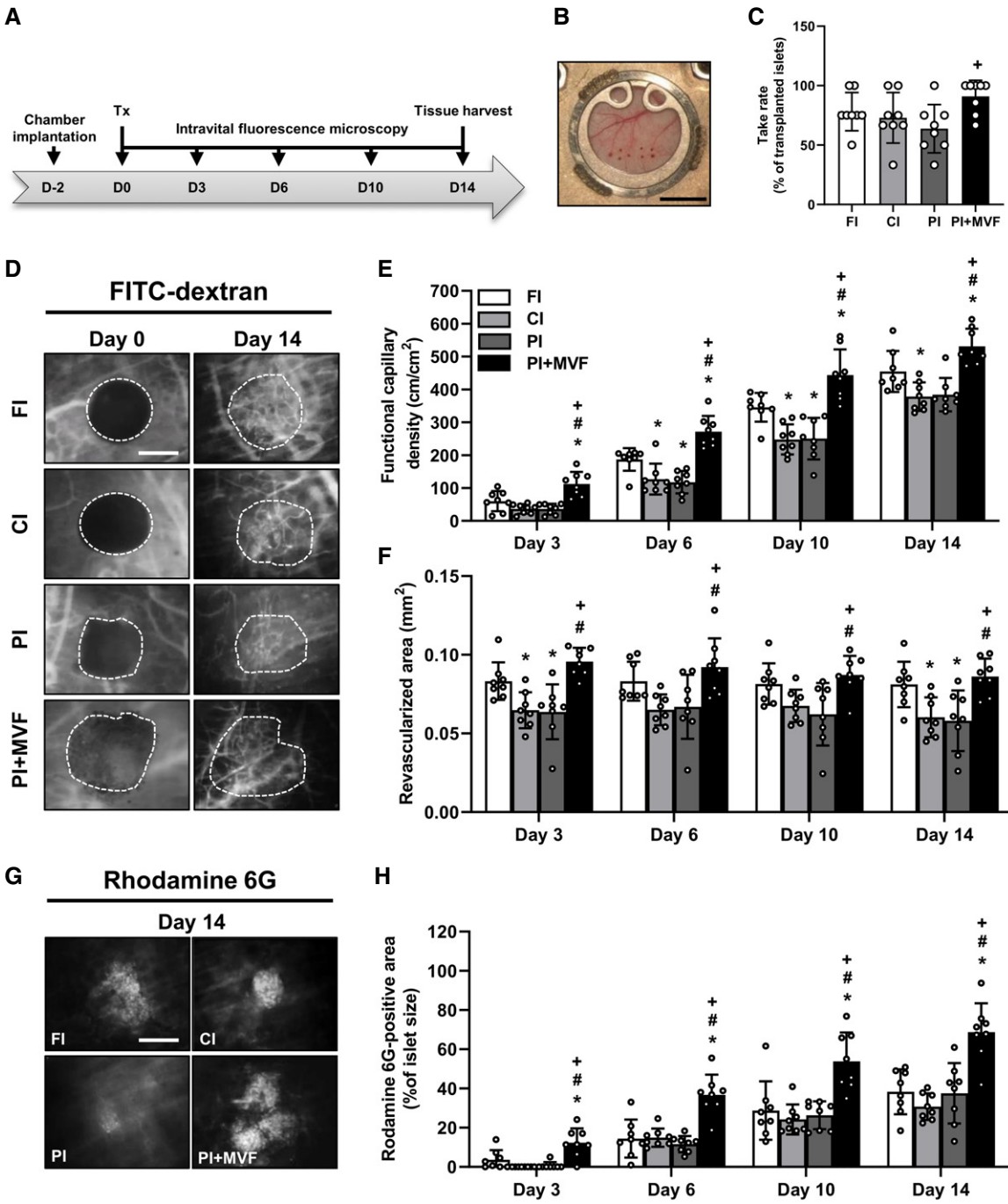

**Figure 6.**

higher number of Ki67-positive cells in PI + MVF on day 3 after transplantation when compared to FI (Fig EV5J). In contrast, we only detected a few casp-3-positive apoptotic cells without any significant differences between the two groups (Fig EV5K). These results indicate that the incorporation of MVF into PI improves the cell proliferation within the islet organoids during the initial post-transplantation phase.

Taken together, our observations demonstrate that the fusion of MVF and pancreatic islet cells to prevascularized islet organoids significantly improves their *in vivo* vascularization, viability, and functionality. Hence, prevascularized islet organoids represent highly promising tissue constructs for clinical islet transplantation.

## Discussion

Pancreatic islet transplantation is a promising strategy for the treatment of T1D; however, the insufficient revascularization of the grafts still represents a major problem. Common vascularization strategies focus on the combination of accessory cells with islets

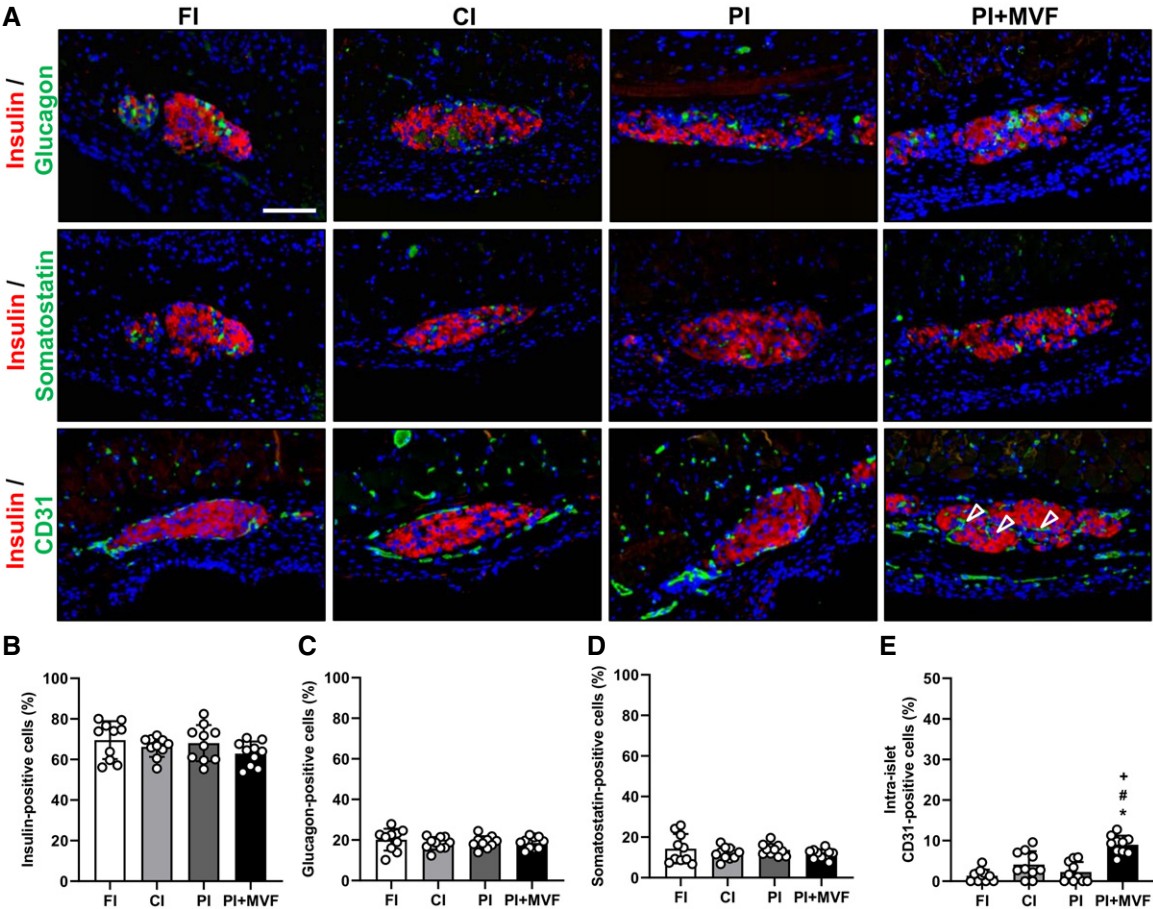

**Figure 7. Cellular composition of transplanted CI, FI, PI, and PI + MVF.**

A       Representative immunofluorescent stainings of insulin/glucagon, insulin/somatostatin and insulin/CD31 in FI, CI, PI, and PI + MVF on day 14 after transplantation. Cell nuclei are stained with Hoechst 33342 (blue). Arrows mark intra-islet endothelial cells. Scale bar: 70 μm.

B–E    Quantitative analysis of insulin- (B), glucagon- (C), somatostatin- (D), and intra-islet CD31-positive cells (E) within FI, CI, PI, and PI + MVF on day 14 after transplantation ($n = 10$ each). Mean ± SD. One-way ANOVA and Tukey's multiple comparisons *post hoc* test were used for statistical analysis. *$P < 0.05$ vs. FI; $^{\#}P < 0.05$ vs. CI; $^{+}P < 0.05$ vs. PI.

Data information: Exact *P*-values are specified in Appendix Table S1.

(Perez-Basterrechea *et al*, 2017) as well as on the incorporation of epithelial and endothelial cells into islet organoids (Penko *et al*, 2011; Lebreton *et al*, 2019; Urbanczyk *et al*, 2020). However, the development of blood vessels out of single cells requires up to 10 days and, thus, cannot prevent hypoxia-induced β-cell death during the initial post-transplantation phase (Korsgren *et al*, 2005). To overcome this problem, we herein introduce a novel concept of islet prevascularization by the fusion of MVF with pancreatic islet cells.

Previous studies reported that the generation of islet organoids containing adipose-derived stem cells (ADSCs) or mesenchymal stem cells (MSCs) results in separated spheroidal units after several days of culture (Jun *et al*, 2014; Shin *et al*, 2015). This indicates that the fusion of pancreatic islet cells with other cell types is difficult. The fusion of MVF and pancreatic islet cells into stable tissue constructs is even more challenging due to the heterogeneous size and the multi-cellular composition of MVF. We found that the successful fusion can be achieved by the combination of 2,000 mouse or human islet cells with 200 MVF. In this context, it should

be noted that pancreatic islets are characterized by a well-defined composition of different endocrine cell types, allowing their communication and interaction for the control of blood glucose levels. Importantly, we could show that the incorporation of MVF does not affect this endocrine function.

It is controversially discussed whether freshly isolated or cultivated islets should be used for clinical islet transplantation. Shapiro *et al* (2000) reported that grafted hFI are capable of inducing normoglycemia in humans. However, it should be considered that for logistic reasons the cultivation of isolated islets for several hours is often hard to circumvent under clinical conditions. Of interest, this seems not to be a major problem. In fact, Froud *et al* (2005) found that the transplantation of hCI even results in an improved clinical outcome when compared to hFI. Based on these findings, we not only used PI but also FI and CI as controls in the present study.

The cultivation of islets may trigger ROS formation, which reduces their cellular viability (Cantley *et al*, 2010; Ma *et al*, 2014). We herein detected an increased number of viable cells within

PI + MVF when compared to controls. Therefore, we speculated that MVF exhibit a protective activity against ROS. This hypothesis was further supported by experiments using isolated islets from mice expressing a mitochondrial ROS sensor for islet organoid generation, which revealed that MVF protect against fusion-induced H₂O₂ formation. To get further insights into the underlying

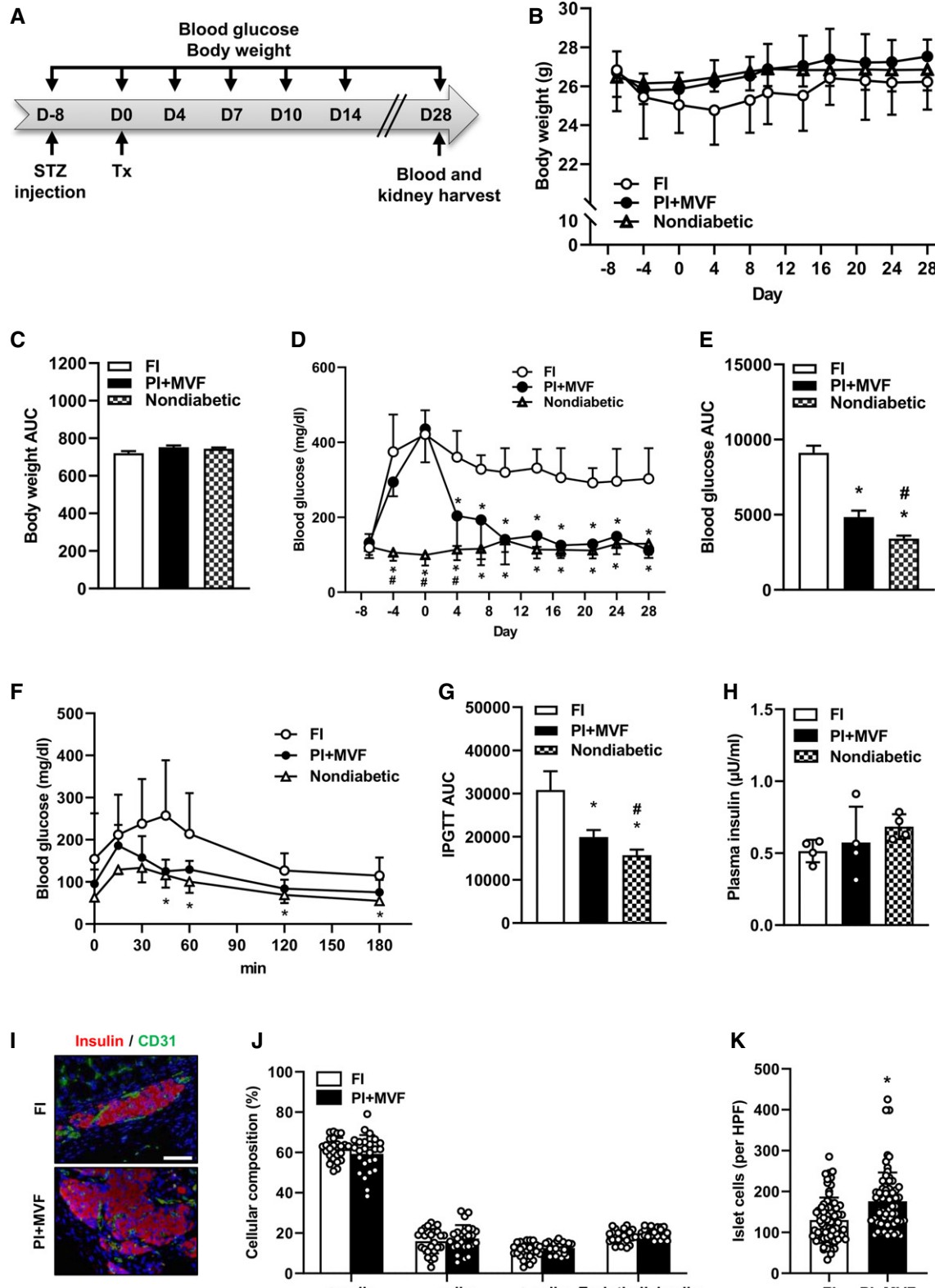

Figure 8.

**Figure 8. *In vivo* endocrine function of CI, FI, PI, and PI + MVF.**

A   Schematic illustration of the experimental setting: Diabetes is induced by a single injection of STZ 8 days before 250 FI or PI + MVF are transplanted under the kidney capsule of diabetic mice ($n = 7$ each). Blood glucose level and body weight are measured from day −8 to day 28 twice a week. On day 28, blood samples and kidney tissue are harvested for immunhistochemical stainings.

B   Quantitative analysis of the body weight (g) of nondiabetic and diabetic mice transplanted with FI or PI + MVF from day −8 to day 28 ($n = 7$ each). Mean ± SD.

C   AUC of the body weight of nondiabetic and diabetic mice transplanted with FI or PI + MVF from day 0 to day 28 ($n = 7$ each). Mean ± SD.

D   Quantitative analysis of blood glucose level (mg/dl) of diabetic mice transplanted with FI or PI + MVF and nondiabetic controls from day −8 to day 28 ($n = 7$ each). Mean ± SD. One-way ANOVA and Tukey's multiple comparisons *post hoc* test were used for statistical analysis. *$P < 0.05$ vs. FI; #$P < 0.05$ vs. PI + MVF.

E   AUC of the blood glucose levels of diabetic mice transplanted with FI or PI + MVF and nondiabetic controls from day 0 to day 28 ($n = 7$ each). Mean ± SD. One-way ANOVA and Tukey's multiple comparisons *post hoc* test were used for statistical analysis. *$P < 0.05$ vs. FI; #$P < 0.05$ vs. PI + MVF.

F   Quantitative analysis of blood glucose level (mg/l) according to the IPGTT of diabetic mice transplanted with FI or PI + MVF and nondiabetic controls to the indicated time points ($n = 5$ each). Mean ± SD. One-way ANOVA and Tukey's multiple comparisons *post hoc* test were used for statistical analysis. *$P < 0.05$ vs. FI.

G   AUC of IPGTT of diabetic mice transplanted with FI or PI + MVF and nondiabetic controls ($n = 5$ each). Mean ± SD. One-way ANOVA and Tukey's multiple comparisons *post hoc* test were used for statistical analysis. *$P < 0.05$ vs. FI; #$P < 0.05$ vs. PI + MVF.

H   Plasma insulin level (μU/ml) of diabetic mice transplanted with FI or PI + MVF and nondiabetic controls at 180 min after IPGTT ($n = 4$ each). Mean ± SD.

I   Representative immunofluorescent stainings of insulin/CD31 in FI and PI + MVF on day 28 after islet transplantation. Scale bar: 50 μm.

J   Quantitative analysis of insulin- (β-cells), glucagon- (α-cells), somatostatin- (δ-cells), and CD31-positive (endothelial) cells within transplanted FI and PI + MVF on day 28 after islet transplantation. Data are expressed in % of total islet cells ($n = 25$). Mean ± SD.

K   Quantitative analysis of islet cells in transplanted FI and PI + MVF per HPF ($n = 70$ each). Mean ± SD. Unpaired Student's *t*-test was used for statistical analysis. *$P < 0.05$ vs. FI.

Data information: Exact *P*-values are specified in Appendix Table S1.

molecular mechanisms, we next analyzed HO-1 expression. We found a markedly increased expression of this enzyme in β-cells of PI + MVF, confirming a strong anti-oxidative activity of MVF in our prevascularized islet organoids. The elevated HO-1 expression may be explained by the hepatocyte growth factor (HGF)/c-Met signaling pathway (Mellado-Gil *et al*, 2011). HGF is secreted by endothelial cells and binds to its transmembrane tyrosine kinase receptor c-Met, which is expressed in β-cells (Dai *et al*, 2005; Roccisana *et al*, 2005). This interaction improves islet graft survival in diabetic animal models (Garcia-Ocana *et al*, 2003). Recently, Chakraborty *et al* (2019) reported that activated c-Met triggers anti-apoptotic protein expression and exerts cytoprotective function against ROS generation via inducing HO-1 expression. Hence, it can be speculated that MVF may induce HO-1 in β-cells by stimulation of the HGF/c-Met pathway.

Our immunohistochemical analysis of PI + MVF did not only show the rapid organization of MVF into dense microvascular networks, but also a significantly higher number of proliferative endothelial cells within prevascularized islet organoids when compared to controls. This observation may be explained by the fact that insulin promotes blood vessel formation via binding to endothelial IR/IGFR (Wang *et al*, 2008; Liu *et al*, 2009; Escudero *et al*, 2017). In line with this finding, we also detected a markedly increased activity of the down-stream proliferating pathways Akt and ERK1/2 in insulin-exposed MVF. Moreover, elevated glucose levels further enhanced the angiogenic sprouting of PI + MVF. Inhibition of IR/IGFR signaling, in turn, suppressed the proliferation of endothelial cells within prevascularized islet organoids.

It should be noted that MVF are not only covered by stabilizing α-SMA-positive cells but also by ASC, a population of pluripotent mesenchymal cells that is predominantly associated with blood vessels (Traktuev *et al*, 2008). In line with this observation, we detected preadipocytes after exposure of PI + MVF to insulin. Traktuev *et al* (2008) reported that the co-transplantation of ASC, endothelial progenitor cells (EPC) and islets improve the formation of neovessels. Furthermore, it is known that ASC protect transplanted islets from inflammatory cytokine-induced cell death (Song *et al*, 2017). Hence, it can be assumed that the herein detected

progenitor cells may further support the angiogenic activity and survival of prevascularized islet organoids.

By means of intravital fluorescence microscopy, we could demonstrate that prevascularized islet organoids exhibit a significantly accelerated and improved blood perfusion after transplantation into dorsal skinfold chambers when compared to controls. Additional immunohistochemical analysis of the grafts revealed increased numbers of intra-islet endothelial cells within PI + MVF, which have already been described to contribute to the revascularization of pancreatic islets (Nyqvist *et al*, 2005). Beside a rapid vascularization, the fenestration of the islet-associated microvasculature is of major importance for endocrine function (Hogan & Hull, 2017). It is well known that VEGF-A triggers microvascular ingrowth and increases blood vessel permeability after islet transplantation (Lammert *et al*, 2003; Brissova *et al*, 2006). Of interest, we detected elevated VEGF-A levels in prevascularized islet organoids, which may have stimulated blood vessel fenestration and improves blood perfusion of transplanted PI + MVF.

Based on these promising *in vitro* and *in vivo* results, we finally analyzed whether the transplantation of prevascularized islet organoids also results in an improved restoration of normoglycemia in STZ-induced diabetic animals. We found that the transplantation of a minimal mass of 250 PI + MVF rapidly normalizes blood glucose levels within 7 days after transplantation, whereas 250 FI did not restore normoglycemia during the entire observation period of 28 days. This result is of major importance considering the fact that the donor to recipient ratio for clinical islet transplantation is still 4:1 to obtain insulin independence (Plesner & Verchere, 2011). Hence, our novel data indicate that the transplantation of prevascularized islet organoids may markedly reduce the number of donors required for this therapeutic approach.

In a future clinical setting, MVF may be isolated from a diabetic patient through enzymatic digestion of aspirated adipose tissue according to standard operating procedures, as already reported for the isolation of adipose tissue-derived stromal-mesenchymal cells (Gimble *et al*, 2010). Because MVF exhibit an improved viability due to shorter enzymatic digestion times and already represent fully

functional vessel segments, they may be even more effective than such single cells (Später *et al*, 2020). The isolation of MVF should be performed on the day of islet delivery to the islet transplant center. Subsequently, MVF and the delivered islet cells may be fused to prevascularized islet organoids in a hospital-associated facility providing good manufacturing practices (GMP) conditions. For this purpose, we introduce in the present study the liquid overlay technique, which allows a precise control of the organoids' size by using defined cell and MVF numbers. However, this technique is time-consuming and, thus, may be difficult to implement into clinical practice. Therefore, it will be necessary to establish novel high-yield fabrication strategies for the generation of prevascularized islet organoids of controlled size and composition. A promising step toward this direction has recently been reported by Wassmer *et al* (2020). They demonstrated that islet organoids can be easily produced by the Sphericalplate 5D™. This plate consists of 12 wells with 750 micro cavities per well, which enables the generation of up to 9,000 organoids per plate. The silicon molds of the plate allow good intercellular contact regardless of the number of cells per microwell, resulting in controlled spheroid sizes and composition.

Taken together, we demonstrate in this study that our novel strategy for the generation of prevascularized islet organoids successfully targets two of the three major obstacles of islet transplantation, that is, (i) scarce donor supply and (ii) failed islet engraftment (Gamble *et al*, 2018). To overcome the third drawback, that is, the need for immunosuppressant therapy to prevent graft rejection (Gamble *et al*, 2018), our strategy may be extended by the fusion of insulin-producing cells originating from human embryonic stem cells (hESC) or induced pluripotent stem cells (iPSC) with autologous MVF obtained from liposuction. Finally, we feel that prevascularization by means of MVF may have also great potential to improve the engraftment and function of other tissue constructs, such as cerebral (Daviaud *et al*, 2018), intestinal (Cortez *et al*, 2018), hepatocyte-like (Pettinato *et al*, 2019), or thymus organoids (Fan *et al*, 2015).

# Materials and Methods

### Reagents and antibodies

All reagents and antibodies used in the present study are listed in Tables EV1–EV3.

### Cell culture

Primary human umbilical vein endothelial cells (HUVEC) and human dermal microvascular endothelial cells endothelial cells (HDMEC) were purchased from PromoCell. The cells were cultivated in Endothelial Cell Basal Medium at 37°C under a humidified 95–5% (v/v) mixture of air and $CO_2$. The cells were passaged at a split ratio of 1:3 after reaching confluence. The cells are regularly tested for mycoplasma contamination.

### Animals

Animals were maintained on a standard 12/12 h day/night cycle. Water and standard pellet chow (Altromin, Lage, Germany) were provided ad libitum. C57BL/6N wild-type and transgenic (Mito-

roGFP2-Orp1) (Fujikawa *et al*, 2016) mice with an age of 8–10 weeks and a body weight of 25–30 g served as donors for islet isolation. Male C57BL/6N wild-type mice and C57BL/6-Tg (CAG-EGFP)1Osb/J mice (purchased from Charles River) with a body weight of 30–35 g served as donors for MVF isolation. Diabetes was induced in 6- to 8-week-old male C57BL/6N wild-type mice.

All experiments were performed according to the German legislation on protection of animals and the National Institutes of Health (NIH) Guide for the Care and Use of Laboratory Animals (Institute of Laboratory Animal Resources, National Research Council, Washington DC, USA). The experiments were approved by the local governmental animal protection committee (permission number: 18/2017 and 45/2018).

### Isolation of pancreatic islets

Mouse pancreatic islets were isolated by collagenase-induced enzymatic digestion and purified by hand picking, as described previously in detail (Li *et al*, 2009). Isolated islets were cultivated in Dulbecco's Modified Eagle Medium (DMEM) supplemented with 10% (v/v) fetal bovine serum (FBS), 100 U/ml penicillin, and 0.1 mg/ml streptomycin at 37°C and 5% $CO_2$ for further experiments.

### Isolation of MVF

Microvascular fragments were isolated by mechanic and enzymatic digestion of epididymal fat pads of mice, as described previously in detail (Frueh *et al*, 2017b). Briefly, mice were anesthetized and the epididymal fat pads were harvested through a midline laparotomy. Subsequently, the fat pads were washed in PBS and digested in collagenase NB4G (0.5 U/ml PBS) for 10 min under vigorous stirring at 37°C. The digestion was stopped by adding DMEM supplemented with 10% (v/v) FBS, 100 U/ml penicillin, and 0.1 mg/ml streptomycin. MVF were separated from the fat layer by gravity and filtration through a 500 μm filter and washed twice by centrifugation at 600 *g* for 5 min at room temperature. We obtained ~ 1 ml fat per mouse, resulting in ~ 40,000 MVF. Finally, the MVF pellet was resuspended in DMEM and the isolated MVF were counted before combining them with islet cells. After isolation, MVF were cultivated in DMEM supplemented with 10% (v/v) FBS, 100 U/ml penicillin, and 0.1 mg/ml streptomycin at 37°C and 5% $CO_2$ for further experiments.

### Generation of PI

Isolated islets were dispersed into single cells by incubation in 0.05% (w/v) trypsin-ethylenediaminetetraacetic acid (EDTA) for 20 s. The enzymatic dispersion was stopped by the addition of DMEM supplemented with 10% (v/v) FBS, 100 U/ml penicillin, and 0.1 mg/ml streptomycin. PI were generated by means of the liquid overlay technique in 96-well plates covered with 1% agarose. The wells were equilibrated with culture medium for 1 h at 37°C prior cell seeding. Murine islets consist of ~ 1,500 cells and human islets of ~ 2,200 cells. In this study, we used 2,000 islet cells for the generation of PI and PI + MVF as well as hPI and hPI + MVF, because cell loss during organoid generation cannot

be excluded. In a next step, 2,000 islet cells were seeded per well and cultivated for 5 days. To form PI + MVF, 2,000 islet cells were cocultured with 200 MVF. The culture medium was changed every second day. After 5 days of cultivation at 37°C and 5% $CO_2$, the organoids were harvested and their diameters were measured by analyzing light microscopic images using FIJI software (NIH, Bethesda, MD, USA).

### Human samples

Islets were isolated from donor organs were obtained with informed written consent and research ethics approval at the University of Alberta (Pro00013094, Pro00001754). Experiments conformed to the principles set out in the WMA Declaration of Helsinki and the Department of Health and Human Services Belmont Report. Human islets were cultured in low-glucose (5.5 mmol/l) DMEM with L-glutamine, 110 mg/l sodium pyruvate, 10% FBS, and 100 units/ml penicillin/streptomycin.

### Immunohistochemistry

Islet organoids, CI and FI (embedded in HepatoQuick® clots), mouse pancreata and kidneys with transplanted islets were fixed for 24 h in 4% paraformaldehyde (PFA) at 4°C. After dehydration, the paraffin-embedded samples were cut into 3-μm-thick sections. The transplanted islets within the kidney capsules were cut sequentially and every 10th slice was immunohistochemically evaluated. Antigens in samples were demasked by citrate buffer and the unspecific binding sites were blocked by goat serum. Cells were stained by incubation with specific primary antibodies against insulin (1:300), glucagon (1:300), somatostatin (1:300), CD31 (1:300), Ki67 (1:200), casp-3 (1:200), and α-SMA (1:500), which were detected by the corresponding secondary antibodies (goat-anti-rabbit-cyanine (Cy)3; goat-anti-mouse-Cy3; goat-anti-rat-Cy3; goat-anti-rat-Alexa Fluor 555, goat-anti-rabbit-streptavidin-peroxidase). Cell nuclei were stained with Hoechst 33342. The sections were analyzed using a BX60F fluorescence microscope (Olympus). The quantification of positively stained cells was done by FIJI software (NIH) and is given in % of all islet cells.

### Mitochondrial $H_2O_2$ sensor detection

Islet organoids, CI, and FI were collected and immediately immersed in phosphate-buffered saline (PBS) containing freshly dissolved 100 mM *N*-ethylmaleimide (NEM) for 20 min at 4°C. Then, the organoids were fixed for 30 min in 4% PFA at 4°C and embedded in HepatoQuick®. After dehydration and paraffin embedding, samples were cut into 3-μm-thick sections and sealed with a cover glass. Microscopic images were taken within 24 h after mounting. Guidelines for the microscopy settings and image analysis of roGFP2-based redox probes have been described previously (Fujikawa *et al*, 2016). Briefly, fluorescence images were acquired by measuring emission at 520–540 nm and exciting sequentially with the 405 and 488 nm laser lines using a 20× objective. For further analysis, images were exported to FIJI software and the ratio between both sensor's images were created by dividing the 405 nm image by the corresponding 488 nm image pixel by pixel. Data are represented as normalized fluorescence ratios (405/488 nm).

### Scanning electron microscopy

The surface topography of islet organoids, CI, and FI was characterized by scanning electron microscopy, as described previously in detail (Laschke *et al*, 2016). Briefly, islet organoids, CI, and FI were washed twice in PBS and fixed in 2% (v/v) glutardialdehyde in 0.1 M sodium cacodylate buffer for 10 min at room temperature (RT). The islets were incubated in osmium tetroxide (1% (v/v) in 0.2 M sodium cacodylate buffer) for 1 h in the dark and dehydrated by incubation in an ethanol series. After sputtering, the dried samples were transferred to conductive carbon adhesive tabs and analyzed in a FEI XL 30 ESEM FEG scanning electron microscopic device (FEI, Hillsboro, OR).

### Western blot analysis

For the preparation of cell lysates, islet organoids, CI, and FI were lysed for 30 min at 4°C in radioimmunoprecipitation assay buffer (Thermo Scientific, Bremen, Germany) containing 0.5 mM phenylmethylsulfonyl fluoride (PMSF) and a protease inhibitor cocktail (1:75 v/v; Sigma-Aldrich). Proteins were separated through a 10% sodium dodecyl sulfate (SDS)–polyacrylamide gel and transferred onto a polyvinylidene fluoride (PVDF) membrane. The membrane was blocked with 5% dry milk in PBS (0.1% Tween 20) for 1 h and then incubated with specific antibodies (anti-CD31 antibody [1:100], anti-α-SMA antibody [1:100], anti-phosphorylated (p)AKT1/2/3 antibody [1:100], anti-AKT antibody [1:500], anti-ERK antibody [1:500], anti-pERK antibody [1:500], anti-HO-1 antibody [1:100], anti-insulin antibody [1:500], anti-GAPDH antibody [1:1,000], or anti-β-actin antibody [1:2,000]). Subsequently, the membrane was washed and incubated for 1 h with the corresponding horseradish peroxidase (HRP)-conjugated secondary antibodies (anti-rabbit antibody [1:1,000], anti-goat antibody [1:2,000], or anti-mouse antibody [1:1,500]). The protein expression was visualized by the incubation of the membrane with enhanced chemoluminescence (ECL) Western blotting substrate (GE Healthcare) in a Chemocam device (Intas). The intensity of the measured signals was quantified using ImageJ software and normalized by the corresponding housekeeping protein.

### Neutral red and trypan blue staining

Islet organoids, CI, and FI (*n* = 20 each) were incubated for 2 min at RT with neutral red (1:100) or trypan blue (1:100) and washed with PBS. The cellular stainings were visualized by bright field images using a 20× objective of BX60 microscope (Olympus).

### Propidium iodide/annexin V and BrdU staining

Islet organoids, CI, FI, and FI cultivated for 24 h in 0.2% $H_2O_2$ were dispersed into single cells by accutase (1:20). Subsequently, the cells were washed in PBS, resuspended in incubation buffer, and stained for 15 min with propidium iodide and annexin V (100 μg/ml), according to the manufacturer's protocol (Roche). The stained cells were analyzed by flow cytometry using FACScan (BD Biosciences) and the amount of vital, apoptotic, necrotic as well as necroptotic cells were given in % of all measured cells.

For BrdU staining, PI + MVF were cultivated for 5 days in liquid overlay technique in the presence of 10 μM BrdU and 800 mM linsitinib or dimethylsulfoxide (DMSO). Afterward, PI + MVF were dispersed and fixed in 70% ethanol for 30 min on ice. The BrdU-FITC stained cells were analyzed by flow cytometry according to the manufacturer's protocol (BD Biosciences).

### Insulin/C-peptide ELISA

The amount of secreted insulin/C-peptide was measured *in vitro* as described previously in detail (Kelly *et al*, 2010). Briefly, 10 islet organoids, CI or FI were washed with Krebs Ringer buffer (KRB; 115 mM NaCl, 4.7 mM KCl, 1.28 mM CaCl$_2$, 1.2 mM MgSO$_4$, 0.1% BSA, 1.1 mM glucose) and incubated for 40 min at 37°C and 5% CO$_2$. The supernatants were collected, and the islets were additionally incubated for 20 min in KRB containing 16.5 mM glucose. The supernatants were collected again, and the amount of secreted insulin was determined by using an insulin or C-peptide ELISA kit according to the manufacturer's protocol. The data were normalized to the corresponding DNA concentration of islets and islet organoids. For this purpose, islets and islet organoids were harvested after glucose-stimulated insulin secretion and the DNA was isolated by using the PureLink Genomic DNA kit according to the manufacturer's protocol. The DNA concentration was measured by means of a NanoDrop. The SI was calculated by dividing the mean DNA normalized insulin values measured from the high and low glucose samples.

### Sprouting assay

Islet organoids, CI, and FI were collected and resuspended in a collagen solution to transfer them into pre-warmed 24-well plates, as previously described (Heiss *et al*, 2015). After 45 min, DMEM supplemented with 10% (v/v) FBS, 100 U/ml penicillin and 0.1 mg/ml streptomycin was added and PI + MVF, PI, CI, and FI were incubated for 72 h at 37°C and 5% CO$_2$. Their sprouting capacity was analyzed by measuring the sprouting area by FIJI software (NIH). Data are given in % of the initial spheroid area (day 0).

### Preparation of the dorsal skinfold chamber and transplantation of PI + MVF, PI, CI, and FI

The dorsal skinfold chamber was prepared, as described previously in detail (Laschke *et al*, 2011). Briefly, two symmetrical titanium frames were implanted on the extended dorsal skinfold of anesthetized mice, resulting in the doubling of the skin in two layers. One layer, including skin, subcutis, and the retractor muscle, was completely removed in a circular area of 15 mm in diameter. This area was then covered by a removable cover slip and snap ring, providing direct microscopic access to the microcirculation of the chamber. After the procedure, the animals were allowed to recover for 72 h.

After recovery, the cover glass was removed and the tissue washed with saline. Subsequently, eight islet organoids, CI, or FI were transplanted onto the exposed striated muscle tissue. Finally, the chamber was sealed with a new cover slip for further intravital fluorescence microscopic analysis.

### Intravital fluorescence microscopy

Dorsal skinfold chamber-equipped mice were anesthetized and received a retrobulbary, intravenous injection of 0.05 ml FITC-labeled dextran (5%) for plasma staining and 0.05 ml rhodamine 6G (2%) for the visualization of blood-perfused and vital endocrine cells (Vajkoczy *et al*, 1995) on day 0 as well as 3, 6, 10, and 14 after islet transplantation. After this injection, the dorsal skinfold chamber was positioned under a fluorescence microscope (Zeiss) with a 100 W mercury lamp attached to a blue (excitation wavelength: 450–490 nm/emission wavelength: > 515 nm) and a green (530–560 nm/> 585 nm) filter block. The microscopic data were recorded by a charge-coupled device video camera (FK6990; Pieper), transferred to a monitor (Trinitron) and a DVD system (DVD-HR775; Samsung) for off-line evaluation.

Microscopic images were analyzed off-line by the computer-assisted image analysis system CapImage (Zeintl, Heidelberg, Germany). The islet size (mm$^2$), the revascularized area (mm$^2$), the functional capillary density (cm/cm$^2$), and the rhodamine 6G-positive area of the islets were assessed as previously described (Menger *et al*, 1992; Ampofo *et al*, 2015). In addition, we measured the diameter (μm), centerline RBC velocity (μm/s), and volumetric blood flow (pl/s) of 4–8 individual microvessels within the grafts (Menger *et al.*, 1992; Ampofo *et al*, 2015). Moreover, we determined the take rate (%), that is, the fraction of engrafted islets in relation to the number of transplanted islets per group on day 14. Successful engraftment was defined as the presence of a newly formed islet-associated blood vessel network.

### Diabetes induction and allogenic islet transplantation

Diabetic phenotypes were induced by a single intraperitoneal injection of 180 mg/kg STZ 8 days prior to islet transplantation. Body weight and non-fasting blood glucose levels of STZ-injected mice were measured twice a week. Blood samples were taken from the tail vein and analyzed by a portable blood glucose monitoring system (GL50; Breuer). Mice with a non-fasting blood glucose level ≥ 350 mg/dl served as recipients for islet transplantation (Sakata *et al*, 2012). Four hundred FI, 300 FI, 250 FI, 200 FI, and 250 PI + MVF were injected under the kidney capsule of diabetic mice using a 10 μl Hamilton syringe, as described previously in detail (Menger *et al*, 2020). Blood glucose levels and body weight were monitored over the entire observation period of 28 days twice a week. Normoglycemia was defined by blood glucose levels below 200 mg/dl.

### IPGTT

Intraperitoneal glucose tolerance test was performed 4 weeks after islet transplantation under the kidney capsule of diabetic mice as previously described (Noguchi *et al*, 2012). Briefly, after 16 h of fasting, the mice were intraperitoneally injected with a 10% glucose solution and their blood glucose levels were determined 0, 15, 30, 45, 60, 120, and 180 min after glucose injection from the tail vein and analyzed by a portable blood glucose monitoring system (GL50; Breuer). Thereafter, the mice were anesthetized and the blood samples were collected from the vena cava. The blood plasma was separated by centrifugation and stored at −80°C. The plasma insulin levels were analyzed by means of an insulin ELISA (Invitrogen, USA).

## Oil Red O staining

Prevascularized islet organoids were seeded on cell culture dishes and cultivated for 5 days in DMEM supplemented with 2 μM insulin, 10% (v/v) FBS, 100 U/ml penicillin, and 0.1 mg/ml streptomycin. Subsequently, the attached organoids were washed with PBS and fixed in 10% formalin at RT for 1 h. Organoids were incubated for 5 min in 60% isopropanol at RT and incubated for 1 h with filtered Oil Red O working solution (60% isopropanol, 8 mg Oil Red O). Bright field images were acquired by means of a BX60 microscope (Olympus) using a 40× objective.

## Real-time quantitative-polymerase chain reaction

Total RNA was isolated using QIAzol lysis reagent (Qiagen). The corresponding cDNA was synthesized from 1 μg of total RNA by QuantiNova Reverse Transcription Kit (Qiagen) according to the manufacturer's instructions. ORA qPCR Green ROX L Mix (highQu) was used for Real-time quantitative-PCR. The data analysis was performed by the MiniOpticon Real-Time PCR System (Bio-Rad). Murine β-actin served as internal control for mRNA detection.

Forward and reverse primers were used in a concentration of 700 nM solved in RNase/DNase-free $H_2O$. Primer sequences for qPCR were coded as follows: VEGF-A forward 5′-GCTGTACCTC CACCATGCCAAG-3′, reverse 5′-CGCACTCCAGGGCTTCATCG-3′; β-actin forward 5′-CCTAGGCACCAGGGTGTGAT-3′, reverse 5′-TCTCC ATGTCGTCCCAGTTG3′.

## Statistical analysis

All *in vitro* experiments were reproduced at least three times. For *in vivo* studies, we used at least five animals per group and no mice were excluded from the statistical analysis. After testing the data for normal distribution and equal variance, differences between two groups were assessed by the unpaired Student's *t*-test. To test differences between multiple groups, one-way ANOVA was applied. This was followed by the Tukey *post hoc* test by means of Prism software 8 (GraphPad). All values are expressed as Mean ± SD. Statistical significance was accepted for $P < 0.05$.

# Data availability

This study includes no data deposited in external repositories.

**Expanded View** for this article is available online.

## Acknowledgements

The authors thank Servier Medical Art for providing access to designed medical elements (https://smart.servier.com/), supporting the generation of graphical items in this publication. We are grateful for the excellent assistance of Caroline Bickelmann, Ruth M. Nickels and Julia Parakenings (Institute for Clinical & Experimental Surgery). This study was supported by an award (Prof. Dr. Theiss-Forschungspreis) of the Freunde des Universitätsklinikums des Saarlandes.

## The paper explained

### Problem

Pancreatic islet transplantation is a promising therapeutic strategy to improve the glycometabolic control in type 1 diabetes mellitus patients. However, a major issue yet to be overcome is the insufficient revascularization of the transplanted islets. Many groups attempt to promote islet revascularization during the initial post-transplantational period by the incorporation or the co-transplantation of endothelial cells. However, the development of new blood vessels from these endothelial cells requires several days and, thus, does not adequately accelerate islet engraftment.

### Result

We generated prevascularized islet organoids by the fusion of fully functional microvessels with pancreatic islet cells. They exhibited a highly angiogenic activity, mediated by a paracrine signaling between β-cells and endothelial cells. More importantly, a critical number of prevascularized islet organoids restored normoglycemia in diabetic animals immediately after transplantation, whereas the identical number of freshly isolated islets did not result in physiological blood glucose levels.

### Impact

Our novel strategy for the generation of prevascularized islet organoids successfully targets two of the three major obstacles of islet transplantation, that is, (i) scarce donor supply and (ii) failed islet engraftment. To overcome the third drawback, that is, the need for immunosuppressant therapy to prevent graft rejection, our strategy may be extended by the fusion of insulin-producing cells originating from human embryonic stem cells (hESC) or induced pluripotent stem cells (iPSC) with autologous MVF obtained from liposuction.

## Author contributions

LN, LPR, MDM, MWL, and EA designed the research and wrote the manuscript. LN, BMS, VB, CK, SW, MP, TS, WM, MMM, FSF, CG, HL, JEMF, PEM, and EA conducted experiments. LN, LPR, WM, HL, PEM, MDM, MWL, and EA analyzed data and interpreted the results. HL, JEMF, and PEM provided human samples and data. EA supervised the study. All authors read and approved the final version of the manuscript.

## Conflict of interest

The authors declare that they have no conflict of interest.

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

transplantation: a methodological comparative study. *Cell Transplant* 29: 963689720937292

