## [Review Process File · EMBO Molecular Medicine]

Improvement of islet transplantation by the fusion of islet cells with functional blood vessels

Lisa Nalbach, Leticia Roma, Beate Schmitt, Vivien Becker, Christina Körbel, Selina Wrublewsky, Mandy Pack, Thomas Später, Wolfgang Metzger, Maximilian Menger, Florian Frueh, Claudia Götz, Haopeng Lin, Jocelyn Manning Fox, Patrick MacDonald, Michael Menger, Matthias W. Laschke, and Emmanuel Ampofo

DOI: [10.15252/emmm.202012616](https://doi.org/10.15252/emmm.202012616)

Corresponding authors: Emmanuel Ampofo (emmanuel.ampofo@uks.eu)

Review Timeline:

Submission Date:	28th Apr 20
Editorial Decision:	25th May 20
Authors' Correspondence:	8th Jun 20
Editor's Email to Reviewer:	15th Jun 20
Editor's Correspondence:	29th Jun 20
Revision Received:	1st Sep 20
Editorial Decision:	18th Sep 20
Revision Received:	25th Sep 20
Accepted:	28th Sep 20

Editor: Lise Roth

Transaction Report:

25th May 2020

Dear Dr. Ampofo,

Thank you for submitting your work to EMBO Molecular Medicine. We have now received feedback from the three reviewers who agreed to evaluate your manuscript. As you will see from the reports below, while referee #1 raises several concerns regarding the experimental execution of the study, referees #2 and #3 are overall positive on the suitability of the manuscript for publication pending adequate revisions.

Given the potential interest of the findings, we will therefore welcome a revised version of your manuscript. Addressing the reviewers' concerns in full will be necessary for further considering the manuscript in our journal, and acceptance of the manuscript will entail a second round of review. EMBO Molecular Medicine encourages a single round of revision only, and acceptance or rejection of the manuscript will depend on the completeness of your responses included in the next, final version of the manuscript. For this reason, and to save you from any frustrations in the end, I would strongly advise against returning an incomplete revision.

When submitting your revised manuscript, please carefully review the instructions that follow below. Failure to include requested items will delay the evaluation of your revision:

- 1) A .docx formatted version of the manuscript text (including legends for main figures, EV figures and tables). Please make sure that the changes are highlighted to be clearly visible.
- 2) Individual production quality figure files as .eps, .tif, .jpg (one file per figure).
- 3) A .docx formatted letter INCLUDING the reviewers' reports and your detailed point-by-point responses to their comments. As part of the EMBO Press transparent editorial process, the point-by-point response is part of the Review Process File (RPF), which will be published alongside your paper.
- 4) A complete author checklist, which you can download from our author guidelines (<https://www.embopress.org/page/journal/17574684/authorguide#submissionofrevisions>). Please insert information in the checklist that is also reflected in the manuscript. The completed author checklist will also be part of the RPF.
- 5) Please note that all corresponding authors are required to supply an ORCID ID for their name upon submission of a revised manuscript.
- 6) Before submitting your revision, primary datasets produced in this study need to be deposited in an appropriate public database (see <https://www.embopress.org/page/journal/17574684/authorguide#dataavailability>). Please remember to provide a reviewer password if the datasets are not yet public.

The accession numbers and database should be listed in a formal "Data Availability " section (placed after Materials & Method). Please note that the Data Availability Section is restricted to new primary data that are part of this study.

7) We would also encourage you to include the source data for figure panels that show essential data. Numerical data should be provided as individual .xls or .csv files (including a tab describing the data). For blots or microscopy, uncropped images should be submitted (using a zip archive if multiple images need to be supplied for one panel). Additional information on source data and instruction on how to label the files are available at .

8) Our journal encourages inclusion of *data citations in the reference list* to directly cite datasets that were re-used and obtained from public databases. Data citations in the article text are distinct from normal bibliographical citations and should directly link to the database records from which the data can be accessed. In the main text, data citations are formatted as follows: "Data ref: Smith et al, 2001" or "Data ref: NCBI Sequence Read Archive PRJNA342805, 2017". In the Reference list, data citations must be labeled with "[DATASET]". A data reference must provide the database name, accession number/identifiers and a resolvable link to the landing page from which the data can be accessed at the end of the reference. Further instructions are available at .

9) We replaced Supplementary Information with Expanded View (EV) Figures and Tables that are collapsible/expandable online. A maximum of 5 EV Figures can be typeset. EV Figures should be cited as 'Figure EV1, Figure EV2" etc... in the text and their respective legends should be included in the main text after the legends of regular figures.

- Additional Tables/Datasets should be labeled and referred to as Table EV1, Dataset EV1, etc. Legends have to be provided in a separate tab in case of .xls files. Alternatively, the legend can be supplied as a separate text file (README) and zipped together with the Table/Dataset file. See detailed instructions here: .

10) The paper explained: EMBO Molecular Medicine articles are accompanied by a summary of the articles to emphasize the major findings in the paper and their medical implications for the non-specialist reader. Please provide a draft summary of your article highlighting

11) For more information: There is space at the end of each article to list relevant web links for further consultation by our readers. Could you identify some relevant ones and provide such

information as well? Some examples are patient associations, relevant databases, OMIM/proteins/genes links, author's websites, etc...

12) Author contributions: the contribution of every author must be detailed in a separate section (before the acknowledgments).

13) Every published paper now includes a 'Synopsis' to further enhance discoverability. Synopses are displayed on the journal webpage and are freely accessible to all readers. They include a short stand first (maximum of 300 characters, including space) as well as 2-5 one-sentences bullet points that summarizes the paper. Please write the bullet points to summarize the key NEW findings. They should be designed to be complementary to the abstract - i.e. not repeat the same text. We encourage inclusion of key acronyms and quantitative information (maximum of 30 words / bullet point). Please use the passive voice. Please attach these in a separate file or send them by email, we will incorporate them accordingly.

Please also suggest a striking image or visual abstract to illustrate your article. If you do please provide a png file 550 px-wide x 400-px high.

14) As part of the EMBO Publications transparent editorial process initiative (see our Editorial at <http://embomolmed.embopress.org/content/2/9/329>), EMBO Molecular Medicine will publish online a Review Process File (RPF) to accompany accepted manuscripts.

In the event of acceptance, this file will be published in conjunction with your paper and will include the anonymous referee reports, your point-by-point response and all pertinent correspondence relating to the manuscript. Let us know whether you agree with the publication of the RPF and as here, if you want to remove or not any figures from it prior to publication.

I look forward to receiving your revised manuscript.

Yours sincerely,

Lise Roth

Lise Roth, PhD
Editor
EMBO Molecular Medicine

To submit your manuscript, please follow this link:

Link Not Available

Photos 400-800 DPI

*Additional important information regarding figures and illustrations can be found at <http://bit.ly/EMBOPressFigurePreparationGuideline>

***** Reviewer's comments *****

Referee #1 (Remarks for Author):

EMM-2020-12616 EMBO Molecular Medicine

Title: Improvement of islet transplantation by the fusion of islet cells with functional blood vessels.

Corresponding Author: Dr. Ampofo

In this study, the authors claimed that they established a novel strategy by fusion of pancreatic islet, they could generate prevascularized islet organoids, and that the prevascularized islet organoids are rapidly blood-perfused after transplantation, so that they could be used to improve the success rates of clinical islet transplantation.

Overall, although the reviewer think that the idea is interesting, however there are many flaws. The authors cocultured islets with microvascular fragments (MVF) derived adipose tissues. In their experimental design, MVF is not labeled, this make it difficult to identify the contribution of MVF. For example, whether the CD31 positive cells are derived from the recipient or the donor should to be shown.

Fig 8 is the main figure. Although the authors claimed that the human islet coculture with MVF showed drastic results, the results are superficial and without proper controls (such as STZ sham control) and in depth analyses, therefore make it difficult to judge. The human insulin secreted from the human islets with or without coculture should be analyzed.

Major critics.

1. Figure 1: The authors describe coculture of 2000 MIN6 cells with 200 MVF from fat pads. The material method should provide information of how many mouse were used for isolating 200 MVF. The authors should mention whether the coculture of human islets was done with mouse MVF or human MVF. If human MVF was used, the site of isolation should be described.

2. Fig 2: The authors claimed that coculture with MVF increased HO-1 expression and reduced ROS. However, they did not show whether the HO1 is expressed in the PI or by the endogenous MVF. The localization of HO1 expression should be examined.
3. Fig 8. MVF seems to give drastic effects to support the islet transplantation. The author should be careful in interpreting their results.
- (i) The author should perform more control experiment to strengthen their findings. For the in vivo transplantation, STZ sham control should be done.
 - (ii) For the in vivo transplantation experiment, the authors should first used mouse islets to show proof of concept before performing the human islets coculture experiments.
 - (iii) The blood insulin levels (mouse/ human insulin) should be evaluated if the levels correlate with the blood glucose levels. The graft should be harvested and examined to confirm the existance of more insulin expressing cells (human derived) in the graft. IPGTT before and after transplantation of each graft experiments should be performed and compared.
 - (iv) The authors failed to identify the donor(s) used for each experiment. Are islets used derived from the same donor(s) in the FI, PI-MVF, CI? This should be mentioned. The authors mentioned that 4 donor derived human islets were used.
 - (v) It is known that human islet show batch differences in their insulin secretion ability. Therefore, insulin secretion activity of islets derived from each donor should be evaluated and be shown.
 - (vi) The authors should examine if the high viability of the transplanted MVF + islets are actually due to the MVF derived vascular cells or the growth factors secreted from MVF. Is the inhibition of islet cell apoptosis or promotion of proliferation is important?
 - (vii) The authors showed spouting of the vascular structure. However, whether or not the blood flow could be detected from the mouse through the human islets should be mentioned.

Referee #2 (Remarks for Author):

This is a nice study, reporting a cell-therapy based strategy to improve islet transplantation outcomes. Authors propose approach which is based on the combination of dissociated islet cells with adipose tissue derived microvascular fragments (MVF). In vitro studies demonstrated that MVF well integrated into the organoids without impairing cellular content and function of spheroids. In vivo studies were performed to demonstrate revascularization and function of the graft. For the vascularization studies, authors used elegant approach to monitor newly growing blood vessels. Organoids (PI-MVF) transplanted into the dorsal skinfold chambers and monitored by intravital fluorescence microscopy showed improved blood perfusion compared to controls. To demonstrate functional potency, marginal mass of PI-MVF were transplanted under the kidney capsule of diabetic mice. In this animal model, transplantation of PI-MVF led to long-term improvement of glycemia, while freshly isolated islets were unable to control glycemia in mice. These claims could have impact in the field of cell-based therapies for diabetes mellitus.

1. Results/ Generation of prevascularized islet organoids: Authors mention that average diameter of the PI+MVF composed of MIN6 cell and MVF was 200 μ m, but they do not give data on size distribution and differences between groups. Please indicate diameters of PI, PI+MVF and whole islets and show differences in cluster sizes between groups if any.
2. Results/ Cellular composition, endocrine function and angiogenic activity of prevascularized islet organoids: According to morphometric data, PI+MVF contain significantly less insulin positive cells compared to other groups. Taking in account that for generation of IP and IP+MVF authors used

same number of islet cells (2000 islet cells/organoid), it seems a little odd. How authors explain this? Please discuss.

3. Results/ Cellular composition, endocrine function and angiogenic activity of prevascularized islet organoids: For insulin secretion tests, please provide additional information on how data were normalized. Additional data on stimulation Index (SI) and the absolute amounts of insulin would have been informative.

4. Results/ Cellular composition, endocrine function and angiogenic activity of prevascularized islet organoids: The authors state that exposure of organoids to high glucose levels (11mM and 25mM) increased angiogenic sprouting from organoids. I find it odd, as physiological fasting blood glucose levels are between 3.9 to 7.1 mmol/L. Exposure of islets and endothelial cells to high glucose concentrations as it happens during diabetes have deleterious effects on these cells and is the main factor of the pathogenesis of the disease. In addition, it has been shown that exposure of endothelial cells to high glucose levels short term impairs their function (<https://doi.org/10.1161/01.CIR.101.22.2618>, <https://doi.org/10.1161/01.CIR.96.1.25>).

5. Results/ Cellular composition, endocrine function and angiogenic activity of prevascularized islet organoids: Please provide further evidence/information on the endothelial nature of the cells constituting sprouts. This could be demonstrated by CD31 staining.

6. Results/ Generation of prevascularized humanized islet organoids: Figure 5 C shows content of endothelial cells in the islets and pseudoislets. Strangely, both freshly isolated islets and cultured islets are completely negative for CD31 staining. Isolated islets contain substantial number of endothelial cells and even contribute to the revascularization of the graft. Please make sure that CD31 antibody is validated on the human derived samples.

7. Materials and methods/ Isolation of MVF: In order to enable replication of experiments please describe the MVF isolation and organoid generation procedure, rather than quoting previous studies

8. Materials and methods/ Neutral red and trypan blue staining: How many islets and organoids were accessed to calculate viability?

9. Materials and methods/ Generation of PI: Please indicate number of cells obtained from one mouse and human islet.

10. Discussion: Authors state: "Common vascularization strategies focus on the incorporation of endothelial cells into pancreatic islets (Lebreton et al, 2019; Pan et al., 2011; Perez Basterrechea et al, 2017)" ... The authors, Lebreton et al, 2019 and Perez Basterrechea et al, 2017 are misquoted. While, Perez Basterrechea et al have used skin derived fibroblasts with whole islets, Lebreton et al reported generation of insulin secreting organoids from islet cells and amniotic epithelial cells. None of them used endothelial cells. Please cite and discuss works of these authors in the context of using accessory/supportive cells to produce hybrid organoids and improve islet engraftment/vascularization.

11. Discussion: Please discuss the translational potential of this approach giving more details on perspectives and challenges, such as organoid generation technique in GMP facility, number of donors that might be needed to generate enough organoids for one recipient ect.

Referee #3 (Remarks for Author):

Nalbach et al describes an approach to pre-vascularize "islet organoids" using microvascular fragments for improved graft survival after transplantation. The authors conclude that

prevascularized "islet organoids" maybe improve clinical islet transplantation success. Overall, the results are interesting and the experiments are performed with the necessary rigor. I have a few suggestions:

- In Fig. 2 the authors suggest that increased HO-1 protects beta cells from oxidative stress, but the WB cannot discriminate if the signal comes from endocrine or endothelial cells. This should be clarified.
- Fig. 3 shows a clear loss of beta cells in PI + MVF. This seems to contradict the results of Fig. 2. Why are beta cells specifically lost after pre-vascularization?
- Fig. 5 shows the pre-vascularization of human islets. The question is if these "islet organoids" survive better in a NOD/Scid mouse model. This would greatly strengthen the translational value, but this is asked for much.
- The authors should at least discuss how feasible human translation is. What would be the source of MVF? How could this be implemented in the routine human islet transplantation protocol, considering the time needed for in vitro pre-vascularization?
- The longer the human islets are in culture, the more beta cells might be lost (see Fig. 3). The authors should discuss pre-vascularization of islets vs pre-vascularization of the transplantation site? e.g. Gamble et al. Shapiro, 2018; Islet. Some transplantation centers try to add stromal-mesenchymal cells for improved graft survival. How easy can MVFs be implemented in such protocols?

Dear Dr. Roth,

we are very pleased that we have the opportunity to resubmit our manuscript "Improvement of islet transplantation by the fusion of islet cells with functional blood vessels" (EMM-2020-12616 EMBO Molecular Medicine) in a revised form. The comments of the reviewers, especially from reviewers 2 and 3 are fair and constructive and we are very optimistic that we will satisfy their comments. We feel, however, that reviewer 1 may have misunderstood our translational approach.

In our study, we first generated mouse islet organoids and analyzed their angiogenic and endocrine function in vitro as a proof of concept. As a next step, we transplanted those mouse islet organoids syngeneically into recipient mice to confirm their improved angiogenic potential and, as a consequence, their increased endocrine function in vivo. To initiate a first step towards the use of this novel concept in clinical practice, we finally generated human islet organoids and analyzed their angiogenic and endocrine function in vitro as a proof of concept for later clinical studies.

We assume that reviewer 1 thought we had transplanted human islet organoids into recipient mice (see major comment 3 (ii) "For the in vivo transplantation experiment, the authors should first used mouse islets to show proof of concept before performing the human islets coculture experiments."). In addition, in several subsequent comments the reviewer recommends more detailed analyses of the in vivo function of transplanted human islets, including analyses of differences between the human islet isolates from different patients and, potentially different centers from all over the world, which, however, may be beyond the scope of the present study (as also mentioned by reviewer 3).

We are not quite sure how to deal with these comments. Would it be possible to have a brief discussion with you on these comments by phone?

Sincerely
Emmanuel

Some time ago, you kindly reviewed the attached manuscript for EMBO Molecular Medicine. Based on the reports from the three reviewers, major revisions of the manuscript were invited.

The authors recently contacted me and had some questions related to your comments (please see document attached).

In particular, there seems to be a misunderstanding regarding the transplantation of murine islet cells with murine microvascular fragments.

Therefore, I would greatly appreciate your feedback on this particular point, and on the authors' proposal to address your concerns.

Thank you very much for your help in the review process,

Sincerely,

Lise Roth

We appreciate the fair and constructive comments of the reviewer 1. We agree with the reviewer that we should provide more information about the isolation procedure of MVF (major comment 1) as well as investigate the expression of HO-1 in more detail (major comment 2). Hence, we are currently performing the recommended experiments as suggested by the reviewer. However, we cannot reply to the subsection (ii, iii, iv, v and vii) of the major comment 3, because there is a fundamental misunderstanding here.

In our study, we generated prevascularized islet organoids by the fusion of **murine** islet cells with **murine** microvascular fragments (MVF). We found that these organoids exhibit a highly angiogenic potential in vitro. Hence, we transplanted the prevascularized **murine** organoids into recipient mice and investigated their revascularization potential as well as their endocrine function. We additionally generated prevascularized islet organoids by the fusion of **human** islet cells and **murine** MVF as a proof of concept. However, it should be noted, that these **humanized** prevascularized islet organoids were solely analyzed in vitro. Although this is described in our manuscript (see page 8, lines 19 and 20), the reviewer seems to believe that we had transplanted the humanized islet organoids into recipient mice. Hence, reviewer 1 requests additional experiments analyzing “the transplanted humanized islets” in more detail. We are aware that the transplantation of islet organoids consisting of human islet cells and murine or even human MVF would greatly strengthen the translational value of our study. However, in line with the comment of reviewer 3, this is out of the scope of the present study.

Due to this misunderstanding, we are not able to reply to the comments concerning transplanted humanized islets (see major comment 3 subsection (ii), (iv), (v) and (vii)). Nonetheless, we will add experiments with prevascularized **murine** islet organoids. For instance, we will perform IPGTT with transplanted murine islet organoids (see major comment 3 (iii)). Moreover, we will include supplementary video material showing the blood flow within the murine grafts (see major comment 3 (vii)).

In order to perform the revision adequately, we would appreciate a timely feedback, if our suggestions are acceptable for reviewer 1.

Dear Emmanuel,

I am getting back to you regarding your question on referee #1's report. This referee has now replied, and you will find his/her report enclosed.

You will see that this referee acknowledges that some points of the initial review were not valid, and thus do not need to be addressed. However, this referee also added a few more points (in red). I would encourage you to address these points as much as possible, but we realize that these comments come long after the initial review, and addressing them all will not be mandatory for further consideration of your manuscript.

Please let me know if you have any question,

With my best wishes,

Lise

EMM-2020-12616 EMBO Molecular Medicine

Title: Improvement of islet transplantation by the fusion of islet cells with functional blood vessels.

Corresponding Author: Dr. Ampofo

In this study, the authors claimed that they established a novel strategy by fusion of pancreatic islet, they could generate prevascularized islet organoids, and that the prevascularized islet organoids are rapidly blood-perfused after transplantation, so that they could be used to improve the success rates of clinical islet transplantation.

Overall, although the reviewer think that the idea is interesting, however there are many flaws. The authors cocultured islets with microvascular fragments (MVF) derived adipose tissues. In their experimental design, MVF is not labeled, this make it difficult to identify the contribution of MVF. For example, whether the CD31 positive cells are derived from the recipient or the donor should to be shown.

Fig 8 is the main figure. Although the authors claimed that the human islet coculture with MVF showed drastic results, the results are superficial and without proper controls (such as STZ sham control) and in depth anaylses, therefore make it difficult to judge. The human insulin secreted from the human islets with or without coculture should be analyzed.

Major critics.

1. Figure 1: The authors describe coculture of 2000 MIN6 cells with 200 MVF from fat pads. The material method should provide information of how many mouse were used for isolating 200 MVF. The authors should mention whether the coculture of human islets was done with mouse MVF or human MVF. If human MVF was used, the site of isolation should be described.
2. Fig 2: The authors claimed that coculture with MVF increased HO-1 expression and reduced ROS. However, they did not show whether the HO1 is expressed in the PI or by the endogenous MVF. The localization of HO1 expression should be examined.
3. Fig 2. In CI, ROS seems to be lower than FI, and comparable with PI+MVF. Although they made some excuse, but the culture This need to be clarified by measuring ROS using other assay method. It seems that HO-1 expression is high, but the ROS reduction is minimum, which does not seem to match the high HO-1 expression.
4. Fig. 3b. The insulin-positive cell % should be lower in PI+MVF since these cells contain the MVF in the mix population. To form PI+MVF, the authors used 2000 islets to cocultured with 200 MVF. Endocrine cell numbers differ between islets, depending on islet size. The authors should note the size range of the islets. How about the MVF? The sizes

- might differ also between different MVF. The authors should mention how they normalize the cell numbers. If the input is the same in different groups, the absolute endocrine cell numbers should be compared.
5. Fig3e, insulin secretion should be normalized to either cell number or DNA amount, etc. not just per 10 islets. The size of islets are very different. The diameter distribution of the islets should be noted.
 6. Figure 5 j: C-peptide secretion is only compared between hFI and hPI+MVF, it is no meaning without comparing with hCI, hPI in addition to hFI and hPI+MV.
 7. Fig 6c The authors did not explain how they assayed for the take rate.
 8. Fig 8. MVF seems to give drastic effects to support the islet transplantation. The author should be careful in interpreting their results.
 - (i) The author should perform more control experiment to strengthen their findings. For the in vivo transplantation, STZ sham control should be done.
 - ~~(ii) For the in vivo transplantation experiment, the authors should first used mouse islets to show proof of concept before performing the human islets coculture experiments.~~
 - (iii) The blood insulin levels (mouse/ human insulin) should be evaluated if the levels correlate with the blood glucose levels. The graft should be harvested and examined to confirm the existence of more insulin expressing cells (human derived) in the graft. IPGTT before and after transplantation of each graft experiments should be performed and compared.
 - ~~(iv) The authors failed to identify the donor(s) used for each experiment. Are islets used derived from the same donor(s) in the FI, PI-MVF, CI? This should be mentioned. The authors mentioned that 4 donor derived human islets were used.~~
 - ~~(v) It is known that human islet show batch differences in their insulin secretion ability. Therefore, insulin secretion activity of islets derived from each donor should be evaluated and be shown.~~
 - (vi) The authors should examine if the high viability of the transplanted MVF + islets are actually due to the MVF derived vascular cells or the growth factors secreted from MVF. Is the inhibition of islet cell apoptosis or promotion of proliferation is important?
 - ~~(vii) The authors showed spouting of the vascular structure. However, whether or not the blood flow could be detected from the mouse through the human islets should be mentioned~~

Review of the manuscript EMM-2020-12616 by Nalbach et al.

We appreciate the fair and constructive comments of the reviewers. In the following, please find our point-by-point reply.

Reply to the comments of reviewer 1

Reviewer comment: In this study, the authors claimed that they established a novel strategy by fusion of pancreatic islet, they could generate prevascularized islet organoids, and that the prevascularized islet organoids are rapidly blood-perfused after transplantation, so that they could be used to improve the success rates of clinical islet transplantation. Overall, although the reviewer think that the idea is interesting, however there are many flaws. The authors cocultured islets with microvascular fragments (MVF) derived adipose tissues. In their experimental design, MVF is not labeled, this make it difficult to identify the contribution of MVF. For example, whether the CD31 positive cells are derived from the recipient or the donor should to be shown.

Reply: According to the comment of the reviewer, we have performed additional experiments to discriminate between CD31-positive endothelial cells derived from the donors' MVF or ingrowing microvessels from the surrounding recipient tissue. For this purpose, we have generated PI+MVF consisting of MVF from GFP-positive mice and islet cells from GFP-negative mice. These PI+MVF were then transplanted into the dorsal skinfold chamber of GFP-negative recipient mice. After 14 days, the grafts were harvested and analyzed by immunohistochemistry. These analyses revealed that 34 ± 13 % of the CD31-positive endothelial cells within the grafts were also positive for GFP. This indicates that a substantial fraction of the microvasculature within the grafts originated from the incorporated MVF. This novel information is now included in the revised version of the manuscript (see page 10, line 28; page 11, lines 1-5; page 17, lines 15 and 16; page 38, lines 3-6; Figure EV4D; marked in yellow).

Reviewer comment: Fig 8 is the main figure. Although the authors claimed that the human islet coculture with MVF showed drastic results, the results are superficial and without proper controls (such as STZ sham control) and in depth analyses, therefore make it difficult to judge. The human insulin secreted from the human islets with or without coculture should be analyzed.

Reply: Please refer to our detailed replies to the reviewer comments 8 i-vi.

Major critics.

Reviewer comment: Figure 1: The authors describe coculture of 2000 MIN6 cells with 200 MVF from fat pads. The material method should provide information of how many mouse were used for isolating 200 MVF. The authors should mention whether the coculture of human islets was done with mouse MVF or human MVF. If human MVF was used, the site of isolation should be described.

Reply: According to the comment of the reviewer, we now provide the information that ~1 mL fat per mouse was obtained, which contained ~40,000 MVF. This indicates that ~ 200 PI+MVF can be generated from one donor mouse. This information is now included in the material and methods section of the revised manuscript (see page 18, lines 12-14; marked in yellow).

In the present study, we used only murine MVF and not human MVF. Therefore, the isolation of human MVF is not described in the material and methods section.

Reviewer comment: Fig 2: The authors claimed that coculture with MVF increased HO-1 expression and reduced ROS. However, they did not show whether the HO1 is expressed in the PI or by the endogenous MVF. The localization of HO1 expression should be examined.

Reply: According to the comment of the reviewer, we have analyzed whether HO-1 is expressed in β -cells or MVF. For this purpose, we have performed additional immunohistochemical and Western blot analyses showing that the expression of HO-1 is mainly restricted to β -cells. This novel information is now included in the results and discussion section of the revised manuscript version (see page 6, lines 14-17; page 13, lines 23-28; page 14, 1-5; page 31, lines 11-14; Fig. 2H and I; marked in yellow).

Reviewer comment: Fig 2. In CI, ROS seems to be lower than FI, and comparable with PI+MVF. Although they made some excuse, but the culture This need to be clarified by measuring ROS using other assay method. It seems that HO-1 expression is high, but the ROS reduction is minimum, which does not seem to match the high HO-1 expression.

Reply: In our experiments, we used transgenic islets isolated from mice expressing the mitochondrial-targeted genetically-encoded roGFP-Orp1 H_2O_2 sensor (Cabezas-Wallscheid *et al.*, 2017; Deglasse *et al.*, 2019; Fujikawa *et al.*, 2016). Using a redox histology approach (Fujikawa *et al.*, 2016), we measured steady-state levels of mitochondrial H_2O_2 in FI, CI, PI and PI+MVF, as shown in Figure 2d and 2e. The genetically-encoded redox sensors are currently the state-of-art methodology to reliably measure differences in H_2O_2 levels (Ezerina *et al.*, 2014; Morgan *et al.*, 2011; Roma & Jonas, 2020; Schwarzlander *et al.*, 2016).

Among several advantages, genetically-encoded roGFP2-based sensors are ratiometric, redox species-specific (in our case, H_2O_2), and allow subcellular localization. The advantages of using these sensors over other small chemical probes and markers have been extensively reviewed and discussed before (Ezerina *et al.*, 2014; Morgan *et al.*, 2011; Roma & Jonas, 2020; Schwarzlander *et al.*, 2016), indicating that chemical probes and oxidative stress markers are non-specific without subcellular localization and prone to artefacts, such as photo-oxidation and uptake differences among treatment.

RoGFP2-based sensors are qualitative and not quantitative, and therefore even small changes in ratio (405/488 nm) can reflect large changes in H_2O_2 levels. For example, dynamic experiments using mito-roGFP2-Orp1 islets show that ratio changes between fully reduced and fully oxidized sensor are on the range of 1.5 ratio units. Addition of 100 μ M of H_2O_2 elicits changes in ratio which are equivalent of 0.5 units. However, this represents an increase of 50% when the degree of sensor oxidation is calculated (Meyer & Dick, 2010; Schwarzlander *et al.*, 2016). Therefore, it is not surprising that steady state differences are not extreme, even though in our case experiments differences were in the range of \sim 0.3 ratio units.

It has been previously shown that roGFP2-Orp1 is mainly reduced by the GSH/GPX system. Therefore, steady state levels are a result of H_2O_2 production versus GSH-mediated scavenging (Meyer & Dick, 2010; Schwarzlander *et al.*, 2016). This might help to explain why increased HO-1 expression does not directly affect sensor oxidation and rather affects its levels indirectly. Therefore, we would not expect a direct correlation of HO-1 levels with sensor oxidation, even though it can be one of the mechanisms by why total H_2O_2 levels are decreased, as suggested in our discussion. Of note, Western blot is also a qualitative rather than quantitative method and therefore little can be concluded about "real" amount of protein expressed.

Finally, the use of transgenic islets expressing H_2O_2 sensor allows us to monitor endogenous changes in mitochondrial H_2O_2 levels that do not depend on incubation of chemicals (such as fluorescent probes and markers) and allow us to only monitor the redox changes specifically

in the islet cells of PI+MVF, as the MVF were isolated from control mice and therefore do not express the sensor.

We therefore feel that using another method with poor specificity, non-ratiometric, and prone to artefacts will bring little additional information to our manuscript.

References

Cabezas-Wallscheid N, Buettner F, Sommerkamp P, Klimmeck D, Ladel L, Thalheimer FB, Pastor-Flores D, Roma LP, Renders S, Zeisberger P *et al* (2017) Vitamin A-Retinoic Acid Signaling Regulates Hematopoietic Stem Cell Dormancy. *Cell* 169: 807-823 e819

Deglasse JP, Roma LP, Pastor-Flores D, Gilon P, Dick TP, Jonas JC (2019) Glucose Acutely Reduces Cytosolic and Mitochondrial H₂O₂ in Rat Pancreatic Beta Cells. *Antioxid Redox Signal* 30: 297-313

Ezerina D, Morgan B, Dick TP (2014) Imaging dynamic redox processes with genetically encoded probes. *Journal of molecular and cellular cardiology* 73: 43-49

Fujikawa Y, Roma LP, Sobotta MC, Rose AJ, Diaz MB, Locatelli G, Breckwoldt MO, Misgeld T, Kerschensteiner M, Herzig S *et al* (2016) Mouse redox histology using genetically encoded probes. *Sci Signal* 9: rs1

Meyer AJ, Dick TP (2010) Fluorescent protein-based redox probes. *Antioxid Redox Signal* 13: 621-650

Morgan B, Sobotta MC, Dick TP (2011) Measuring E(GSH) and H₂O₂ with roGFP2-based redox probes. *Free Radic Biol Med* 51: 1943-1951

Roma LP, Jonas JC (2020) Nutrient Metabolism, Subcellular Redox State, and Oxidative Stress in Pancreatic Islets and beta-Cells. *J Mol Biol* 432: 1461-1493

Schwarzlander M, Dick TP, Meyer AJ, Morgan B (2016) Dissecting Redox Biology Using Fluorescent Protein Sensors. *Antioxid Redox Signal* 24: 680-712

Reviewer comment: Fig. 3b. The insulin-positive cell % should be lower in PI+MVF since these cells contain the MVF in the mix population. To form PI+MVF, the authors used 2000 islets to cocultured with 200 MVF. Endocrine cell numbers differ between islets, depending on islet size. The authors should note the size range of the islets. How about the MVF? The sizes might differ also between different MVF. The authors should mention how they normalize the cell numbers. If the input is the same in different groups, the absolute endocrine cell numbers should be compared.

Reply: We agree with the reviewer that the endocrine cell number differs between individual islets depending on their size. To minimize the variability of cell numbers in individual islets, we used in our study murine and human FI and CI in a narrow size range of ~160-170 μ m. The reviewer is correct that individual MVF also differ in their length. In fact, most MVF used in this study exhibited a length of ~10-60 μ m. Accordingly, they also contained varying cell numbers. Based on these differences of individual cell numbers between the different groups, we have decided to normalize the data of the cellular composition to the overall number of cells within FI, CI, PI and PI+MVF. This information is now included in the results section of the revised manuscript (see page 5, lines 11-13; page 6, lines 23-28; page 7, lines 1-4, page 9, lines 5-11; page 36, lines 14-20; Figure EV2A-C; marked in yellow).

Reviewer comment: Fig3e, insulin secretion should be normalized to either cell number or DNA amount, etc. not just per 10 islets. The size of islets are very different. The diameter distribution of the islets should be noted.

Reply: According to the comment of the reviewer, we have normalized the insulin and C-peptide secretion to the DNA amount of the individual groups (see page 7, lines 9-14; page

9, lines 15-16; page 22, lines 11-17; page 31, lines 24-27; page 33, lines 18-19; page 36, lines 21-25; page 37, lines 20-23; Fig. 3F; Fig. 5J; Figure EV2D and E; Figure EV3F and G; marked in yellow).

Reviewer comment: Figure 5 j: C-peptide secretion is only compared between hFI and hPI+MVF, it is no meaning without comparing with hCI, hPI in addition to hFI and hPI+MV.

Reply: We agree with the reviewer that the C-peptide secretion from the additional control groups (hCI and hPI) would complete the data set. However, the current situation (COVID-19 pandemic and the lockdown of routine clinical care) markedly affects the islet isolation procedure and shipping of human islets from clinical islet transplant centers. Therefore, delays in delivery are to be expected, which would have a significant impact on islet function and viability in our experiments. Moreover, our in vivo experiments in the dorsal skinfold chamber model clearly showed that FI exhibited the highest angiogenic activity of our control groups (FI, CI and PI) (Fig. 6E). Accordingly, we feel that hFI are also the most relevant control group for the analysis of C-peptide secretion. Therefore, we have decided not to perform additional analyses of C-peptide secretion from hCI and hPI for the revised version of the manuscript.

Reviewer comment: Fig 6c The authors did not explain how they assayed for the take rate.

Reply: According to the comment of the reviewer, we now provide a detailed description in the revised manuscript how we calculated the take rate. This reads as follows:

'Moreover, we determined the take rate (%), i.e. the fraction of engrafted islets in relation to the number of transplanted islets per group on day 14. Successful engraftment was defined as the presence of a newly formed islet-associated blood vessel network.'

(see page 23, lines 28; page 24, lines 1 and 2; marked in yellow)

Reviewer comment: Fig 8. MVF seems to give drastic effects to support the islet transplantation. The author should be careful in interpreting their results.

(i) The author should perform more control experiment to strengthen their findings. For the in vivo transplantation, STZ sham control should be done.

Reply: According to the comment of the reviewer, we have included an additional STZ sham control group. As expected, we found pathological blood glucose levels in non-transplanted STZ-treated mice during the entire observation period of 28 days. This information is now included in the revised version of the manuscript (see page 11, lines 11-13; page 38, lines 13-23, Figure EV5B-F; marked in yellow).

~~*(ii) For the in vivo transplantation experiment, the authors should first used mouse islets to show proof of concept before performing the human islets coculture experiments.*~~

Reply: This comment has been retracted by the reviewer after discussion with the editor.

~~*(iii) The blood insulin levels (mouse/ human insulin) should be evaluated if the levels correlate with the blood glucose levels. The graft should be harvested and examined to confirm the existence of more insulin expressing cells (human derived) in the graft. IPGTT*~~

before and after transplantation of each graft experiments should be performed and compared.

Reply: According to the comment of the reviewer, we have performed an additional IPGTT before and after transplantation and we have determined the plasma insulin levels 180 min after IPGTT. This information is now included in the revised version of the manuscript, which reads as follows:

'An intraperitoneal glucose tolerance test (IPGTT) demonstrated that the blood glucose levels of mice transplanted with PI+MVF were lower when compared to animals transplanted with FI, resulting in a significantly reduced AUC (Fig 8F and G). In contrast, the plasma insulin levels, which were determined at 180 min after glucose injection, did not differ between the groups (Fig 8H).'

(see page 11, lines 21-26; page 24, lines 16-24; page 35, lines 12-18; page 38, lines 17-23; Fig. 8F-H; Figure EV5D-F; marked in yellow).

In the original version of our manuscript, we had already analyzed the overall number of islet cells on day 28. We detected a significantly higher overall number of islet cells in mice transplanted with PI+MVF when compared to those receiving FI. In addition, we detected no differences in the fraction of insulin-, glucagon-, somatostatin- and CD31-expressing cells between the two groups. Taken together, these findings demonstrate that PI+MVF finally contain a higher absolute number of insulin-expressing β -cells when compared to FI (see page 12, lines 3-8; Fig. 8J and K; marked in yellow).

~~*(iv) The authors failed to identify the donor(s) used for each experiment. Are islets used derived from the same donor(s) in the FI, PI-MVF, CI? This should be mentioned. The authors mentioned that 4 donor derived human islets were used.*~~

Reply: This comment has been retracted by the reviewer after discussion with the editor.

~~*(v) It is known that human islet show batch differences in their insulin secretion ability. Therefore, insulin secretion activity of islets derived from each donor should be evaluated and be shown.*~~

Reply: This comment has been retracted by the reviewer after discussion with the editor.

~~*(vi) The authors should examine if the high viability of the transplanted MVF + islets are actually due to the MVF derived vascular cells or the growth factors secreted from MVF. Is the inhibition of islet cell apoptosis or promotion of proliferation is important?*~~

Reply: According to comment of the reviewer, we have additionally investigated whether the high viability of PI+MVF is caused by cell proliferation or apoptosis. For this purpose, we transplanted additional FI and PI+MVF under the kidney capsule of diabetic mice. The grafts were harvested on day 3 and analyzed by means of Ki67- and cleaved caspase-3 (casp-3) stainings. Of interest, we detected a significantly higher number of Ki67-positive cells in grafted PI+MVF on day 3. In contrast, we only detected a few casp-3-positive apoptotic cells without any significant differences between the two groups. These results indicate that the incorporation of MVF into PI improves the cell proliferation within the islet organoids during the initial post-transplantation phase. This information is now included in the revised version

of our manuscript (see page 12, lines 3-10; page 19, line 20; page 38, line 28; page 39, lines 1-4; Figure EV5I-K; marked in yellow).

~~(vii) The authors showed spouting of the vascular structure. However, whether or not the blood flow could be detected from the mouse through the human islets should be mentioned.~~

Reply: This comment has been retracted by the reviewer after discussion with the editor.

Reply to the comments of reviewer 2

Reviewer comment: Results/ Generation of prevascularized islet organoids: Authors mention that average diameter of the PI+MVF composed of MIN6 cell and MVF was 200 μm , but they do not give data on size distribution and differences between groups. Please indicate diameters of PI, PI+MVF and whole islets and show differences in cluster sizes between groups if any.

Reply: According to the comment of the reviewer, we now provide the diameters of PI, PI+MVF and whole islets of the different groups in the revised version of our manuscript (see page 5, lines 11-13; page 9, lines 5-6; page 36, lines 14-19; Figure EV2A and B; marked in yellow).

Reviewer comment: Results/ Cellular composition, endocrine function and angiogenic activity of prevascularized islet organoids: According to morphometric data, PI+MVF contain significantly less insulin positive cells compared to other groups. Taking in account that for generation of IP and IP+MVF authors used same number of islet cells (2000 islet cells/organoid), it seems a little odd. How authors explain this? Please discuss.

Reply: In our immunohistochemical analysis of the cellular composition of FI, CI, PI and PI+MVF we did not show the absolute number of insulin-positive cells but the fraction of insulin-positive cells in % of all islet cells. This is due to the fact that the PI solely consisted of 2,000 islet cells, whereas the PI+MVF consisted of 2,000 islet cells and 200 MVF. Accordingly, PI+MVF consisted of much more non-endocrine cells than PI, resulting in a lower fraction of insulin-positive cells. In addition, it should be noted that individual MVF differ in their length. In fact, most MVF used in this study exhibited a length of $\sim 10\text{-}60\ \mu\text{m}$. Accordingly, they also contained varying cell numbers. Based on these differences of individual cell numbers between the different groups, we have decided to normalize the data of the cellular composition to the overall number of cells within FI, CI, PI and PI+MVF. This information is now included in the results section of the revised manuscript (see page 6, lines 23-28; page 7, lines 1-4; page 36, lines 14-20; Figure EV2A-C; marked in yellow).

Reviewer comment: Results/ Cellular composition, endocrine function and angiogenic activity of prevascularized islet organoids: For insulin secretion tests, please provide additional information on how data were normalized. Additional data on stimulation Index (SI) and the absolute amounts of insulin would have been informative.

Reply: In the original manuscript, we had provided the absolute amount of insulin secretion in $\mu\text{U}/\text{mL}$, because the islet organoids differed in their cell number due to the incorporated MVF (see also comment above). However, we agree with the reviewer that the normalized data and the SI would still have been informative. Hence, we have replaced the absolute by the normalized data of insulin secretion in the revised version of our manuscript. The original data on absolute insulin secretion and the SI are now provided in the supplementary data part of the revised manuscript version (see page 7, lines 9-14; page 9, lines 15-16; page 22, lines 11-17; page 31, lines 24-27; page 33, lines 18-19; page 36, lines 21-25; page 37, lines 20-23; Fig. 3F; Fig. 5J; Figure EV2D and E; Figure EV3F and G; marked in yellow).

Reviewer comment: Results/ Cellular composition, endocrine function and angiogenic activity of prevascularized islet organoids: The authors state that exposure of organoids to high glucose levels (11mM and 25mM) increased angiogenic sprouting from organoids. I find it odd, as physiological fasting blood glucose levels are between 3.9 to 7.1 mmol/L. Exposure of islets and endothelial cells to high glucose concentrations as it happens during diabetes

have deleterious effects on these cells and is the main factor of the pathogenesis of the disease. In addition, it has been shown that exposure of endothelial cells to high glucose levels short term impairs their function (<https://doi.org/10.1161/01.CIR.101.22.2618>, <https://doi.org/10.1161/01.CIR.96.1.25>).

Reply: We agree with the reviewer that several studies demonstrated negative effects of high glucose levels on the function of endothelial cells. However, in general it should be noted that the findings of studies focusing on endothelial cell function in diabetes are still quite contradictory. In fact, it is well known that hyperglycemia in diabetic patients is typically associated with excessive angiogenesis in different organs, which, for example, causes diabetic retinopathy and nephropathy (Aouiss *et al*, 2019; Zent & Pozzi, 2007). Moreover, several in vitro studies reported that high glucose-treated endothelial cells exhibit an increased proliferating, migrating and tube forming activity due to the up-regulation of angiogenic signaling pathways (Betts-Obregon *et al*, 2016; Fernando *et al*, 2018; Qiu *et al*, 2018; Shi *et al*, 2019). Furthermore, we have recently shown that the exposition to high glucose (30 mM) for 24 h results in a significantly higher number of proliferating endothelial and perivascular cells and a markedly improved in vivo vascularization capacity of MVF (Laschke *et al*, 2019). Hence, we feel that all these previous studies are in line with our novel results on the angiogenic activity of prevascularized islet organoids in high-glucose medium.

References

Aouiss A, Anka Idrissi D, Kabine M, Zaid Y (2019) Update of inflammatory proliferative retinopathy: Ischemia, hypoxia and angiogenesis. *Current research in translational medicine* 67: 62-71

Betts-Obregon BS, Vellanki S, Buikema J, Tsin AT, Wright K (2016) Effect of Glucose on Retinal Endothelial Cell Viability and VEGF Secretion. *HSOA journal of cell biology & cell metabolism* 3

Fernando KHN, Yang HW, Jiang Y, Jeon YJ, Ryu B (2018) Diploretrohydroxycarmalol Isolated from *Ishige okamurae* Represses High Glucose-Induced Angiogenesis In Vitro and In Vivo. *Mar Drugs* 16

Laschke MW, Seifert MS, Scheuer C, Kontaxi E, Metzger W, Menger MD (2019) High glucose exposure promotes proliferation and in vivo network formation of adipose-tissue-derived microvascular fragments. *Eur Cell Mater* 38: 188-200

Qiu F, Tong H, Wang Y, Tao J, Wang H, Chen L (2018) Recombinant human maspin inhibits high glucose-induced oxidative stress and angiogenesis of human retinal microvascular endothelial cells via PI3K/AKT pathway. *Mol Cell Biochem* 446: 127-136

Shi Y, Chen C, Xu Y, Liu Y, Zhang H, Liu Y (2019) LncRNA FENRR promotes high-glucose-induced proliferation and angiogenesis of human retinal endothelial cells. *Biosci Biotechnol Biochem* 83: 869-875

Zent R, Pozzi A (2007) Angiogenesis in diabetic nephropathy. *Seminars in nephrology* 27: 161-171

Reviewer comment: Results/ Cellular composition, endocrine function and angiogenic activity of prevascularized islet organoids: Please provide further evidence/information on the endothelial nature of the cells constituting sprouts. This could be demonstrated by CD31 staining.

Reply: As suggested by the reviewer, we have performed additional immunohistochemical analyses to assess the endothelial nature of cells sprouting out of PI+MVF. Of interest, we found that the sprouts indeed stained positive for the endothelial cell marker CD31. This novel finding is now included in the revised version of our manuscript (see page 8, lines 18-19; page 32, lines 24-27; Fig. 4L; marked in yellow).

Reviewer comment: Results/ Generation of prevascularized humanized islet organoids: Figure 5 C shows content of endothelial cells in the islets and pseudoislets. Strangely, both

freshly isolated islets and cultured islets are completely negative for CD31 staining. Isolated islets contain substantial number of endothelial cells and even contribute to the revascularization of the graft. Please make sure that CD31 antibody is validated on the human derived samples.

Reply: In our study, we incorporated murine MVF into human islet organoids. Therefore, we used in Fig. 5c a mouse-specific CD31 antibody to detect mouse endothelial cells. This antibody does not recognize human CD31. Therefore, we could not detect any endothelial cells in hFI and hCI in our original analysis. However, we agree with the reviewer that hFI should contain a substantial number of human endothelial cells. To visualize these cells, we have additionally performed stainings with a human-specific CD31 antibody. The results of these stainings are now included in the revised version of our manuscript (see page 9, lines 7-11; page 37, lines 16-19; Figure EV3D and E; marked in yellow). They show - as expected - a substantial number of endothelial cells within hFI.

Reviewer comment: Materials and methods/ Isolation of MVF: In order to enable replication of experiments please describe the MVF isolation and organoid generation procedure, rather than quoting previous studies.

Reply: According to the comment of the reviewer, we now provide a detailed description of the isolation procedure of MVF (see page 18, lines 6-14; marked in yellow). A detailed description of the organoid generation procedure has already been given in the original version of our manuscript (see page 18, lines 18-28; page 19, lines 1 and 2).

Reviewer comment: Materials and methods/ Neutral red and trypan blue staining: How many islets and organoids were accessed to calculate viability?

Reply: Neutral red and trypan blue staining was performed in 20 FI, 20 CI, 20 PI, 20 PI+MVF and 20CI + H₂O₂. This information is given now in the revised version of our manuscript (see page 21, line 15; marked in yellow)

Reviewer comment: Materials and methods/ Generation of PI: Please indicate number of cells obtained from one mouse and human islet.

Reply: According to the comment of the reviewer, we have additionally analyzed the cell number of murine and human FI after their dissociation into single cells. We found that murine islets (mean diameter: ~160-170 µm) consist of ~ 1,500 cells and human islets (mean diameter: ~165-180 µm) of ~ 2,200 cells. This information is now included in the material and methods section of the revised manuscript version (see page 18, lines 24-26; marked in yellow).

Reviewer comment: Discussion: Authors state: "Common vascularization strategies focus on the incorporation of endothelial cells into pancreatic islets (Lebreton et al, 2019; Pan et al., 2011; Perez Basterrechea et al, 2017)"... The authors, Lebreton et al, 2019 and Perez Basterrechea et al, 2017 are misquoted. While, Perez Basterrechea et al have used skin derived fibroblasts with whole islets, Lebreton et al reported generation of insulin secreting organoids from islet cells and amniotic epithelial cells. None of them used endothelial cells. Please cite and discuss works of these authors in the context of using accessory/supportive cells to produce hybrid organoids and improve islet engraftment/ vascularization.

Reply: We thank the reviewer for drawing attention to this citation error. According to the suggestion of the reviewer, we have cited the findings of Lebreton et al, 2019 and Perez Basterrechea et al, 2017 in the right context and we have added new references of (Penko et al, 2011; Urbanczyk et al, 2020). These authors incorporated endothelial cells as well as endothelial progenitor cells into islet organoids to improve islet engraftment and vascularization (see page 11, lines 18-21; marked in yellow).

References

Penko D, Mohanasundaram D, Sen S, Drogemuller C, Mee C, Bonder CS, Coates PT, Jessup CF (2011) Incorporation of endothelial progenitor cells into mosaic pseudoislets. *Islets* 3: 73-79

Urbanczyk M, Zbinden A, Layland SL, Duffy G, Schenke-Layland K (2020) Controlled Heterotypic Pseudo-Islet Assembly of Human beta-Cells and Human Umbilical Vein Endothelial Cells Using Magnetic Levitation. *Tissue engineering Part A* 26: 387-399

Reviewer comment: Discussion: Please discuss the translational potential of this approach giving more details on perspectives and challenges, such as organoid generation technique in GMP facility, number of donors that might be needed to generate enough organoids for one recipient ect.

Reply: According to the comment of the reviewer, we have included a novel paragraph in the revised version of our manuscript, which discusses strategies to translate our novel approach for the generation of prevascularized islet organoids into clinical practice. This paragraph reads as follows:

‘In a future clinical setting, MVF may be isolated from a diabetic patient through enzymatic digestion of aspirated adipose tissue according to standard operating procedures, as already reported for the isolation of adipose tissue-derived stromal-mesenchymal cells (Gimble et al, 2010). Because MVF exhibit an improved viability due to shorter enzymatic digestion times and already represent fully functional vessel segments, they may be even more effective than such single cells (Spater et al, 2020). The isolation of MVF should be performed on the day of islet delivery to the islet transplant center. Subsequently, MVF and the delivered islet cells may be fused to prevascularized islet organoids in a hospital-associated facility providing Good Manufacturing Practices (GMP) conditions. For this purpose, we introduce in the present study the liquid overlay technique, which allows a precise control of the organoids’ size by using defined cell and MVF numbers. However, this technique is time-consuming and, thus, may be difficult to implement into clinical practice. Therefore, it will be necessary to establish novel high-yield fabrication strategies for the generation of prevascularized islet organoids of controlled size and composition. A promising step towards this direction has recently been reported by Wassmer et al. (Wassmer et al, 2020). They demonstrated that islet organoids can be easily produced by the Sphericalplate 5D™. This plate consists of 12 wells with 750 micro cavities per well, which enables the generation of up to 9000 organoids per plate. The silicon molds of the plate allow good intercellular contact regardless of the number of cells per microwell, resulting in controlled spheroid sizes and composition.’

(see page 15, lines 19-28; page 16; lines 1-9; marked in yellow).

References

Gimble JM, Guilak F, Bunnell BA (2010) Clinical and preclinical translation of cell-based therapies using adipose tissue-derived cells. *Stem Cell Res Ther* 1: 19

Spater T, Tobias AL, Menger MM, Nickels RM, Menger MD, Laschke MW (2020) Biological coating with platelet-rich plasma and adipose tissue-derived microvascular fragments improves the vascularization, biocompatibility and tissue incorporation of porous polyethylene. *Acta biomaterialia* 108: 194-206

Wassmer CH, Bellofatto K, Perez L, Lavallard V, Cottet-Dumoulin D, Ljubicic S, Parnaud G, Bosco D, Berishvili E, Lebreton F (2020) Engineering of Primary Pancreatic Islet Cell Spheroids for Three-dimensional Culture or Transplantation: A Methodological Comparative Study. *Cell transplantation* 29: 963689720937292

Reply to the comments of reviewer 3

Reviewer comment: In Fig. 2 the authors suggest that increased HO-1 protects beta cells from oxidative stress, but the WB cannot discriminate if the signal comes from endocrine or endothelial cells. This should be clarified.

Reply: According to the comment of the reviewer, we have analyzed whether HO-1 is expressed in β -cells or MVF. For this purpose, we have performed additional immunohistochemical and Western blot analyses showing that the expression of HO-1 is mainly restricted to β -cells. This novel information is now included in the results and discussion section of the revised manuscript version (see page 6, lines 14-17; page 13, lines 23- 28; page 14, lines 1-5; page 31, lines 11-14; Fig. 2H and I; marked in yellow)

Reviewer comment: Fig. 3 shows a clear loss of beta cells in PI + MVF. This seems to contradict the results of Fig. 2. Why are beta cells specifically lost after pre-vascularization?

Reply: In our immunohistochemical analysis of the cellular composition of FI, CI, PI and PI+MVF we did not show the absolute number of insulin-positive cells but the fraction of insulin-positive cells in % of all islet or islet organoid cells. This is due to the fact that the PI solely consisted of 2,000 islet cells, whereas the PI+MVF consisted of 2,000 islet cells and additional 200 MVF. Accordingly, PI+MVF consisted of much more non-endocrine cells than PI, resulting in a lower fraction of insulin-positive cells. In addition, it should be noted that individual MVF differ in their length. In fact, most MVF used in this study exhibited a length of ~10-60 μm . Accordingly, they also contained varying cell numbers. Based on these differences of individual cell numbers between the different groups, we have decided to normalize the data of the cellular composition to the overall number of cells within FI, CI, PI and PI+MVF. This information is now included in the results section of the revised manuscript (see page 6, lines 23-28; page 7, lines 1-4; page 36, lines 14-20; Figure EV2A-C; marked in yellow)

Reviewer comment: Fig. 5 shows the pre-vascularization of human islets. The question is if these "islet organoids" survive better in a NOD/Scid mouse model. This would greatly strengthen the translational value, but this is asked for much.

Reply: We agree with the reviewer that the transplantation of human prevascularized islet organoids into immunodeficient mice would strengthen the impact of our study. However, due to the COVID-19 pandemic it is currently very difficult for us to receive additional human islets from our cooperation partners. In addition, it would be necessary to isolate MVF from human adipose tissue for this translational approach. This, however, has still to be established. For these reasons, we agree with the reviewer that the suggested approach is beyond the scope of the present study. Nonetheless, we plan to take this next important step towards the clinical translation of our approach in a future follow-up study.

Reviewer comment: The authors should at least discuss how feasible human translation is. What would be the source of MVF? How could this be implemented in the routine human islet transplantation protocol, considering the time needed for in vitro pre-vascularization?

Reply: According to the comment of this reviewer and another comment of reviewer 2, we have included a novel paragraph in the revised version of our manuscript, which discusses strategies to translate our novel approach for the generation of prevascularized islet organoids into clinical practice. This paragraph reads as follows:

'In a future clinical setting, MVF may be isolated from a diabetic patient through enzymatic digestion of aspirated adipose tissue according to standard operating procedures, as already

reported for the isolation of adipose tissue-derived stromal-mesenchymal cells (Gimble *et al*, 2010). Because MVF exhibit an improved viability due to shorter enzymatic digestion times and already represent fully functional vessel segments, they may be even more effective than such single cells (Spater *et al*, 2020). The isolation of MVF should be performed on the day of islet delivery to the islet transplant center. Subsequently, MVF and the delivered islet cells may be fused to prevascularized islet organoids in a hospital-associated facility providing Good Manufacturing Practices (GMP) conditions. For this purpose, we introduce in the present study the liquid overlay technique, which allows a precise control of the organoids' size by using defined cell and MVF numbers. However, this technique is time-consuming and, thus, may be difficult to implement into clinical practice. Therefore, it will be necessary to establish novel high-yield fabrication strategies for the generation of prevascularized islet organoids of controlled size and composition. A promising step towards this direction has recently been reported by Wassmer *et al*. (Wassmer *et al*, 2020). They demonstrated that islet organoids can be easily produced by the Sphericalplate 5D™. This plate consists of 12 wells with 750 micro cavities per well, which enables the generation of up to 9000 organoids per plate. The silicon molds of the plate allow good intercellular contact regardless of the number of cells per microwell, resulting in controlled spheroid sizes and composition.'

(see page 15, lines 19-28; page 16; lines 1-9; marked in yellow).

References

Gimble JM, Guilak F, Bunnell BA (2010) Clinical and preclinical translation of cell-based therapies using adipose tissue-derived cells. *Stem Cell Res Ther* 1: 19

Spater T, Tobias AL, Menger MM, Nickels RM, Menger MD, Laschke MW (2020) Biological coating with platelet-rich plasma and adipose tissue-derived microvascular fragments improves the vascularization, biocompatibility and tissue incorporation of porous polyethylene. *Acta biomaterialia* 108: 194-206

Wassmer CH, Bellofatto K, Perez L, Lavallard V, Cottet-Dumoulin D, Ljubicic S, Parnaud G, Bosco D, Berishvili E, Lebreton F (2020) Engineering of Primary Pancreatic Islet Cell Spheroids for Three-dimensional Culture or Transplantation: A Methodological Comparative Study. *Cell transplantation* 29: 963689720937292

Reviewer comment: The longer the human islets are in culture, the more beta cells might be lost (see Fig. 3). The authors should discuss pre-vascularization of islets vs pre-vascularization of the transplantation site? e.g. Gamble et al. Shapiro, 2018; Islet. Some transplantation centers try to add stromal-mesenchymal cells for improved graft survival. How easy can MVFs be implemented in such protocols?

Reply: According to the comment of the reviewer, we have included a novel paragraph in the Introduction section of our revised manuscript, which discusses the prevascularization of the islet transplantation site. This paragraph reads as follows:

'A promising strategy to achieve this is the prevascularization of the transplantation site, as already described for subcutaneous tissue by means of a catheter technique (Pepper *et al*, 2015). However, in this case the revascularization of the islet grafts is crucially dependent on the angiogenic ingrowth of new microvessels from the surrounding host tissue, which is still a time-consuming process.'

(see page 3, lines 11-16; marked in yellow).

References

Pepper AR, Gala-Lopez B, Pawlick R, Merani S, Kin T, Shapiro AM (2015) A prevascularized subcutaneous device-less site for islet and cellular transplantation. *Nat Biotechnol* 33: 518-523

In addition, we have included a novel paragraph in the Discussion section of our revised manuscript, which addresses the implementation of MVF isolation protocols in protocols for the incorporation of stromal-mesenchymal cells into islet grafts to improve graft survival (see also our reply to reviewer comment 4).

18th Sep 2020

Dear Dr. Ampofo,

Thank you for the submission of your revised manuscript to EMBO Molecular Medicine. We have now received the enclosed reports from the two referees who reviewed the new version of your manuscript. As you will see, they are now supportive of publication, and I am thus pleased to inform you that we will be able to accept your manuscript pending the following final minor amendments:

1) Main manuscript text:

- Please answer/correct the changes suggested by our data editors in the main manuscript file (in track changes mode). This file will be sent to you in the next couple of days. Please use this file for any further modification.
- Please remove the highlighted text.
- Material and methods: Please indicate whether the cells were checked for mycoplasma contamination. Please indicate the source of the mice. For human samples, include a statement that informed consent was obtained from all subjects and that the experiments conformed to the principles set out in the WMA Declaration of Helsinki and the Department of Health and Human Services Belmont Report (this should also be mentioned in the checklist).
- In the figure legends, please indicate the exact p= values, not a range, along with the statistical test used. Some people found that to keep the figures clear, providing a supplemental table with all exact p-values was preferable. You are welcome to do this if you want to.
- Funding: please ensure that complete funding information is listed.

2) The paper explained: EMBO Molecular Medicine articles are accompanied by a summary of the articles to emphasize the major findings in the paper and their medical implications for the non-specialist reader. Please provide a draft summary of your article highlighting

3) For more information: There is space at the end of each article to list relevant web links for further consultation by our readers. Could you identify some relevant ones and provide such information as well? Some examples are patient associations, relevant databases, OMIM/proteins/genes links, author's websites, etc...

4) Figures:

Fig EV2 A-E are referenced before Fig EV1 C. in the main text, and Fig EV 2 F,G are referenced after Fig EV3 A-C. Please modify to have the figures referenced in the chronological order.

5) Thank you for providing Source Data. Please upload them so as to have 1 pdf file per figure. Please make sure that your figures within the manuscript are uploaded with high resolution.

6) Checklist:

Please fill section E, human subjects.

7) Every published paper now includes a 'Synopsis' to further enhance discoverability. Synopses are displayed on the journal webpage and are freely accessible to all readers. They include a short stand first (maximum of 300 characters, including space) as well as 2-5 one-sentences bullet points that summarize the paper. Please write the bullet points to summarize the key NEW findings. They should be designed to be complementary to the abstract - i.e. not repeat the same text. We encourage inclusion of key acronyms and quantitative information (maximum of 30 words / bullet point). Please use the passive voice. Please attach these in a separate file or send them by email, we will incorporate them accordingly.

8) As part of the EMBO Publications transparent editorial process initiative (see our Editorial at <http://embomolmed.embopress.org/content/2/9/329>), EMBO Molecular Medicine will publish online a Review Process File (RPF) to accompany accepted manuscripts.

In the event of acceptance, this file will be published in conjunction with your paper and will include the anonymous referee reports, your point-by-point response and all pertinent correspondence relating to the manuscript. Let us know whether you agree with the publication of the RPF and as here, if you want to remove or not any figures from it prior to publication.

I look forward to receiving your revised manuscript.

Yours sincerely,

Lise Roth

Lise Roth, PhD
Editor
EMBO Molecular Medicine

To submit your manuscript , please follow this link:

Link Not Available

The system will prompt you to fill in your funding and payment information. This will allow Wiley to send you a quote for the article processing charge (APC) in case of acceptance. This quote takes into account any reduction or fee waivers that you may be eligible for. Authors do not need to pay any fees before their manuscript is accepted and transferred to our publisher.

***** Reviewer's comments *****

Referee #1:

The manuscript is suitable for publication.

Referee #2:

The manuscript is suitable for publication.

The authors performed the requested editorial changes.

28th Sep 2020

Dear Dr. Ampofo,

We are pleased to inform you that your manuscript is now accepted for publication and is being sent to our publisher to be included in the next available issue of EMBO Molecular Medicine.

We would like to remind you that as part of the EMBO Publications transparent editorial process initiative, EMBO Molecular Medicine will publish a Review Process File online to accompany accepted manuscripts. If you do NOT want the file to be published or would like to exclude figures, please immediately inform the editorial office via e-mail.

Please read below for additional important information regarding your article, its publication and the production process.

Congratulations on your interesting work!

With my best wishes,

Lise

Lise Roth, Ph.D
Editor
EMBO Molecular Medicine

Follow us on Twitter @EmboMolMed
Sign up for eTOCs at embopress.org/alertsfeed

*** ** IMPORTANT INFORMATION ** **

SPEED OF PUBLICATION

The journal aims for rapid publication of papers, using using the advance online publication "Early View" to expedite the process: A properly copy-edited and formatted version will be published as "Early View" after the proofs have been corrected. Please help the Editors and publisher avoid delays by providing e-mail address(es), telephone and fax numbers at which author(s) can be contacted.

Should you be planning a Press Release on your article, please get in contact with embomolmed@wiley.com as early as possible, in order to coordinate publication and release dates.

LICENSE AND PAYMENT:

All articles published in EMBO Molecular Medicine are fully open access: immediately and freely available to read, download and share.

EMBO Molecular Medicine charges an article processing charge (APC) to cover the publication costs. You, as the corresponding author for this manuscript, should have already received a quote with the article processing fee separately. Please let us know in case this quote has not been received.

Once your article is at Wiley for editorial production you will receive an email from Wiley's Author Services system, which will ask you to log in and will present you with the publication license form for completion. Within the same system the publication fee can be paid by credit card, an invoice, pro forma invoice or purchase order can be requested.

Payment of the publication charge and the signed Open Access Agreement form must be received before the article can be published online.

PROOFS

You will receive the proofs by e-mail approximately 2 weeks after all relevant files have been sent to our Production Office. Please return them within 48 hours and if there should be any problems, please contact the production office at embopressproduction@wiley.com.

Please inform us if there is likely to be any difficulty in reaching you at the above address at that time. Failure to meet our deadlines may result in a delay of publication.

All further communications concerning your paper proofs should quote reference number EMM-2020-12616-V3 and be directed to the production office at embopressproduction@wiley.com.

Corresponding Author Name: Emmanuel Ampofo

Manuscript Number: EMM-2020-12616